# ON THE EFFECT OF SAMPLING DIVERSITY IN SCALING LLM INFERENCE

## ABSTRACT

Large language model (LLM) scaling inference is key to unlocking greater performance, and leveraging diversity has proven an effective way to enhance it. Motivated by the observed relationship between solution accuracy and meaningful response diversity, we systematically study the effect of prompt diversity in scaling inference. We theoretically explain **why** diversified sampling improves Best-of-$N$ scaling, showing that responses generated from diverse prompts after Best-of-$N$ selection exhibit significantly lower error rates than those produced from stationary prompts. Building on this analysis, we derive a diversity-fidelity trade-off principle, that guides the design of sampling strategies introducing diversity. From this guidance, we instantiate a family of effective perturbation styles. We theoretically and empirically characterize **when** diversified exploration remains effective, demonstrating that it works under a variety of conditions, and we further show that under majority voting, diversity may vanish. Finally, we systematically evaluate how effective sampling diversity is and show that, when applied appropriately in different contexts, it yields relative gains of 10.8% in EM@100 for reasoning, 9.6% for mathematics, and 9.5% in Pass@100 for code generation. Overall, this work provides a systematic analysis that offers a theoretical and empirical foundation for understanding how sampling diversity affects LLM inference-time scaling.

## 1 INTRODUCTION

Large language models (LLMs) have shown impressive performance across diverse tasks. As their capabilities grow, studying and improving their inference processes becomes increasingly crucial. LLM scaling inference is known to exhibit non-determinism, with variability arising from stochastic decoding, floating-point precision limits, and system-level concurrency (Yuan et al., 2025; Atil et al., 2024). Recent work has sought to eliminate this variability: (Yuan et al., 2025) traced accuracy drops to precision-induced rounding differences and advocated higher-precision inference, while (He & Lab, 2025) intro-

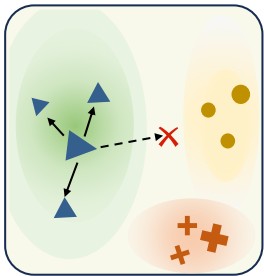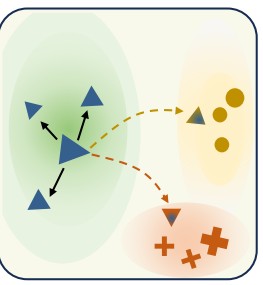

(a) Direct Sampling    (b) Diversified Sampling

Figure 1: A brief sketch of (a) direct sampling without diversification and (b) diversified sampling.

duced batch-invariant kernels to stabilize GPU scheduling. However, such nondeterminism can be beneficial for test-time scaling (Wang et al., 2023b; Li et al., 2025; 2023a; Mu et al., 2024; Naik et al., 2023; Zeng et al., 2024; Wu et al., 2024; Nori et al., 2024; Snell et al., 2024; Brown et al., 2024; Gandhi et al., 2024; Snell et al., 2025; Lee et al., 2025; Wang et al., 2025a), where model parameters remain fixed, and performance improvements must therefore arise from encouraging diverse and thus non-deterministic exploration. Previous literature has investigated temperature scaling as a means to increase output diversity and, in turn, improve downstream task performance, while other studies (Li et al., 2023a; Mu et al., 2024; Naik et al., 2023; Hu et al., 2025) have focused on designing diversified prompts to improve inference accuracy in reasoning tasks, eliciting varied

| Strategies | Pass@100 | tf-idf sim. | BERT sim. | lev. sim. | seq. sim. |
|---|---|---|---|---|---|
| None | 0.8212 | 0.2152 | 0.9997 | 0.2219 | 0.2244 |
| Role | 0.8262 | 0.2063 | 0.9996 | 0.2161 | 0.2208 |
| Instruction | 0.8992 | 0.1625 | 0.9968 | 0.1767 | 0.1884 |
| Jabberwocky | 0.8161 | 0.2173 | 0.9997 | 0.2267 | 0.2285 |

Table 1: **Effects of different injection strategies.** 100 solutions were generated using `gpt-4o-mini` (OpenAI, 2023b) for each strategy on the MBPP benchmark.

chain-of-thought intermediate steps that guide the model's reasoning process toward convergent answers through consensus.

However, Best-of-$N$ sampling (Cobbe et al., 2021; Lightman et al., 2023), as a typical form of test-time scaling, aims to maximize the utility of pre-trained models by efficiently exploring multiple responses and selecting the most accurate one. In such cases, sampling solutions from an LLM using the same prompt often leads to similar outputs, "trapped" within a local cluster (Figure 1(a)). The concentrated nature of the trapped solutions might stem from the limited diversity imposed by post-training objectives, which are typically designed to optimize zero-shot performance and align LLM as instruction-following chatbots (Xiang et al., 2025). These objectives often prioritize optimizing the model to produce a single, correct answer, which mismatches with the goal of repeated sampling. The commonly used distillation technique may also diminish model diversity (Cideron et al., 2024; DeepSeek-AI et al., 2025). Diverse candidate solutions should span multiple clusters, with responses distributed across a broader solution space, breaking out of local clusters (Figure 1(b)). An intuitive strategy is to introduce diversity at the prompt level, which in turn broadens the exploration space. We conducted fundamental empirical studies within the Best-of-$N$ framework by applying diversified prompt perturbations, as shown in Table 1. The diversity strategies employed to promote varied responses include *Role* and *Instruction* prompt perturbations, described in Section 4.2 as task-level approaches. These are compared against direct sampling (*None*) and *Jabberwocky*, the latter introducing a grammatically well-formed but irrelevant text. The **Pass@k** rate measures the proportion of correct solutions among $k$ generated attempts in code generation tasks, whereas **tf-idf**, **BERT**, **Levenshtein**, and **token sequence** measure the similarity between solutions (see Appendix C for details). Table 1 shows that the pass rate improves when perturbation strategies produce candidate solutions with reduced similarity. *This observation motivates us to investigate the effect of exploration diversity on scaling inference.*

In this paper, we systematically study the effect of diversified sampling. We first ❶ provide a theoretical perspective showing why general exploration diversity improves best-of-$N$ performance. We then ❷ examine the effect of perturbation fidelity, ❸ instantiate several perturbation styles, and ❹ study their behavior under different conditions. Extensive evaluations on reasoning, mathematics, and code generation show that the perturbations yeilds relative gains of 10.8% in EM@100 for reasoning, 9.6% for mathematics, and 9.5% in Pass@100 for code generation compared to direct sampling. In summary, we address our contributions as follows:

★ We theoretically show why exploration diversity, possibly achieved through perturbation-induced prompt variation, can improve Best-of-$N$ performance. The results suggest that increasing sampling diversity leads to faster convergence as $N \to \infty$ and a lower asymptotic error.

★ Building upon our main theorem, we derive a diversity-fidelity tradeoff principle, serving as a guideline to choose sampling strategies that introduce diversity while preserving fidelity. Empirical evidence supports the theoretical implication that moderately relevant perturbations improve performance, while overly similar or irrelevant ones yield no performance gains and can even lead to degradation. Guided by the this, we instantiate a group of potentially effective perturbation styles.

★ We theoretically and empirically analyze when diversified exploration remains effective, showing that sampling diversity is robust across varying temperatures, the presence of CoT steps, and LLM-as-a-Judge verification, while its benefits still depend on the thinker model's strength and the perturbation cardinality. At the same time, diversity cannot be applied indiscriminately: our theoretical analysis identifies majority voting as a failure mode where diversity does not lead to performance increase, which we also confirm empirically.

★ We systematically evaluate diversified sampling on reasoning, math, and code-generation tasks, showing that, when applied appropriately in different contexts, task-level and query-level perturbations, as well as their combination, can improve Best-of-N scaling to varying degrees.

## 2 PRELIMINARIES

**Task Description.** We consider sets of tasks defined by a tuple $\langle \boldsymbol{p}, \mathcal{Q}, V \rangle$ of an instruction prompt $\boldsymbol{p}$, a distribution $\mathcal{Q}$ over the question set and a verifier $V$. For a solver of the task, the **prompt** $\boldsymbol{p}$ and a **question** $\boldsymbol{q}$ sampled from the distribution $\mathcal{Q}(\cdot)$ are given, from which the solver predicts an **answer** $\boldsymbol{s}$. This answer is finally judged by the **verifier** $V(\boldsymbol{s}|\boldsymbol{p}, \boldsymbol{q})$, which assigns 1 to accepted answers and 0 to rejected answers. In **reasoning** and **math** tasks, the prompt $\boldsymbol{p}$ asks the solver to choose answer $\boldsymbol{s}$ from an answer set $\boldsymbol{A}$ for some question $\boldsymbol{q} \sim \mathcal{Q}$, and the verifier $V$ checks if the answer exactly matches the hidden ground truth. In **code generation** task, the solver is given a prompt and object pair $\langle \boldsymbol{p}, \boldsymbol{o} \rangle$ in natural language with $\boldsymbol{o} \sim \mathcal{Q}$, which asks the solver to write code for some object $\boldsymbol{o}$. The objective is to implement $\boldsymbol{o}$ such that it passes all hidden tests evaluating its correctness. A solution $s'$ is deemed correct by the verifier $V$ if it successfully passes all hidden tests.

**Best-of-N sampling.** Repeatedly sampling i.i.d. responses $[\boldsymbol{s}]_N := [\boldsymbol{s}_1, \boldsymbol{s}_2, ..., \boldsymbol{s}_N] \sim \text{LLM}(\cdot|\boldsymbol{p}, \boldsymbol{q})$ given prompt $\boldsymbol{p}$ and question $\boldsymbol{q}$ from the LLM solver. For reasoning and math tasks, a task is considered to be solved if at least one response exactly matches the ground truth (Wang et al., 2023a); in this case the proportion of tasks that are solved by the LLM solver with $k$ attempts is called the **EM@k rate**. For code generation tasks, a task is solved if at least one attempt passes all hidden tests (this is equivalent to selecting the answer that passes the highest number of validation tests (Chen et al., 2024a)); in this case the proportion of tasks that are solved with $k$ attempts is called the **Pass@k rate** (Chen et al., 2021). More details on evaluation metrics can be found in Appendix C.

## 3 WHY SOLUTION DIVERSITY IMPROVES PERFORMANCE: A THEORETICAL PERSPECTIVE

To understand why solution diversity enhances best-of-N performance, we provide a theoretical perspective as its justification. For a more technically rigid description and the proof of our theorem, please refer to Appendix B.

**Setting.** First, we formalize the settings of our theories by introducing some useful notations. We use $\boldsymbol{r} = [\boldsymbol{p}, \boldsymbol{q}]$ to denote concatenated inputs to the LLM. To characterize different sampling strategies, we configure each attempt by a *decoding mode* $\zeta \in \mathcal{Z}$ (e.g., decoding seed/sampler, temperature, reasoning style), and define a *diversity policy* $\nu$ as a distribution over modes. Given $\boldsymbol{r}$ and $\zeta \sim \nu$, the *LLM solver* produces $\boldsymbol{s} \sim \text{LLM}(\cdot \mid \boldsymbol{r}, \zeta)$ and a *verifier* $V$ assigns a $0/1$ correctness score. We define the per–mode failure log–probability to be

$$q(\boldsymbol{r}, \zeta) := \log \mathbb{P}_{\boldsymbol{s} \sim \text{LLM}(\cdot|\boldsymbol{r}, \zeta)}\big[V(\boldsymbol{s}) = 0\big]. \tag{1}$$

**Hybrid diversity.** We factor the diversified attempt mode as $\zeta = (\eta, \xi)$, where $\eta \sim \nu_0$ is the *base* randomness within decoding configuration (e.g. a suite of different temperatures to be used during sampling), while $\xi \sim \Pi$ is an *auxiliary* diversity source (e.g. prompt perturbation); we write interchangeably $q(\boldsymbol{r}, \eta, \xi) = q(\boldsymbol{r}, \zeta)$. The goal is to prove this additional diversity boosts performance.

**Hypotheses.** To quantitatively capture the positive effect of diversified sampling on performance, we now posit two hypotheses—*dispersion* and *fidelity*—to characterize the basic properties of diversified sampling, in preparation for our main results.

**Hypothesis 3.1** (Variation under auxiliary diversity). Fix an input $\boldsymbol{r}$ for which there exists $\zeta = (\eta, \xi)$ with $q(\boldsymbol{r}, \zeta) \neq 0$. Define the first absolute central moment with respect to varying $\xi$:

$$M_1(\boldsymbol{r}; \nu_0, \Pi) := \mathbb{E}_{\eta \sim \nu_0} \mathbb{E}_{\xi \sim \Pi} \big| q(\boldsymbol{r}, \eta, \xi) - \mathbb{E}_{\xi \sim \Pi} q(\boldsymbol{r}, \eta, \xi) \big|, \tag{2}$$

then there exists a constant $\hat{\mu}_1 > 0$ such that $M_1(\boldsymbol{r}; \nu_0, \Pi) \geq \hat{\mu}_1$.

*Remark* 3.2 (Intuition for Hypothesis 3.1). Unless $\boldsymbol{r}$ is truly unsolvable for *all* $(\eta, \xi)$, varying the auxiliary randomness $\xi$ induces variation into the success probability. This is because a large auxiliary diversity (e.g. prompt perturbation) can lead to aggressively different decoding paths compared to base policy, which can then lead to larger variations in success probabilities. Note that averaging over $\eta$ reflects that we compare attempts under the same baseline stochasticity.

**Hypothesis 3.3** (Fidelity with fixed auxiliary randomness)**.** There exist $\epsilon \in [0, 1)$ such that, when fixed, any auxiliary setting $\xi^\star$ used by the diversified policy performs almost as well as the baseline reference policy $\xi^0$ for Best-of-$N$ pass rate for $N \geq 1$:

$$\frac{\mathbb{E}_{\boldsymbol{r} \sim \mathcal{R}} \, \mathbb{E}_{\eta \sim \nu_0} \big[ \exp \big\{ N \, q(\boldsymbol{r}, \eta, \xi^\star) \big\} \big]}{\mathbb{E}_{\boldsymbol{r} \sim \mathcal{R}} \, \mathbb{E}_{\eta \sim \nu_0} \big[ \exp \big\{ N \, q(\boldsymbol{r}, \eta, \xi^0) \big\} \big]} \; \leq \; 1 + \epsilon. \tag{3}$$

*Remark* 3.4 (Intuition for Hypothesis 3.3)**.** Here the auxiliary choice $\xi$ is *fixed* on both sides so that both policies enjoy the same amount of base variability ($\eta \sim \nu_0$). Fidelity requires that switching to a new auxiliary setting $\xi^\star$ does not globally worsen the response quality compared to a reference $\xi^0$. It is worth noting that Hypothesis 3.3 should be considered a guideline for selecting good auxiliary diversity sources rather than an assumption: good decoding modes should preserve the fidelity of generation results. For example, with prompt perturbation, one should first test if a specific perturbation method degrades generation quality before applying it as part of the diversity suite.

**Main result.** Finally, with the hypotheses ready, we compare the base diversity policy of $(\eta, \xi_0) \sim \nu_0 \times \{\xi_0\}$ and hybrid diversity policy of $(\eta, \xi) \sim \nu = \nu_0 \times \Pi$. We write Best-of-$N$ failure probabilities as

$$P_{\text{div}}^N \; := \; \mathbb{P}\Big[ V(\boldsymbol{s}_k) = 0, \; \forall k \in [N] \; \Big| \; \boldsymbol{s}_k \sim \text{LLM}(\cdot \mid \boldsymbol{r}, \zeta_k), \; \zeta_k \overset{\text{i.i.d.}}{\sim} \nu, \; \boldsymbol{r} \sim \mathcal{R} \Big], \tag{4}$$

$$P_{\text{reg}}^N \; := \; \mathbb{P}\Big[ V(\boldsymbol{s}_k) = 0, \; \forall k \in [N] \; \Big| \; \boldsymbol{s}_k \sim \text{LLM}(\cdot \mid \boldsymbol{r}, \zeta), \; \zeta \sim \nu_0 \times \{\zeta_0\}, \; \boldsymbol{r} \sim \mathcal{R} \Big]. \tag{5}$$

As $N \to \infty$, these converge to limits $P_{\text{div}}^{\text{inf}}$ and $P_{\text{reg}}^{\text{inf}}$, which reflect the fractions of inputs that remain unsolved under the respective policies.

**Theorem 3.5** (Diversity improves Best-of-$N$)**.** *Under Hypotheses 3.1 and 3.3, there exists a positive sequence $C_N = \Omega\big(\hat{\mu}_1^2 N / (1 + \epsilon)\big)$, increasing in $N$, such that*

$$P_{\text{div}}^N \; \leq \; \big(P_{\text{reg}}^N - P_{\text{reg}}^{\text{inf}}\big) / (1 + C_N) \; + \; P_{\text{div}}^{\text{inf}}, \qquad \text{with} \quad P_{\text{div}}^{\text{inf}} \; \leq \; P_{\text{reg}}^{\text{inf}}. \tag{6}$$

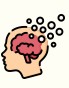

> *This theorem implies two distinct advantages of introducing auxiliary diversity: (i) **Lower asymptote.** Diversity shrinks the "blind-spot" set of instances that remain unsolved as $N$ grows. (ii) **Faster convergence.** The error reduction factor improves with richer (but faithful) diversity, yielding steeper Best-of-$N$ gains.*

## 4 TOWARDS ENCOURAGING EXPLORATION DIVERSIFICATION

In this section, following our theoretical insights, we study prompt perturbation as a means to enhance solution diversity. We begin by deriving a diversity–fidelity trade-off principle, which serves as a guideline for choosing sampling perturbations. This principle is then corroborated by our empirical observations. Guided by it, we instantiate a set of perturbation strategies that are likely to be effective.

### 4.1 PATTERNS OF PERTURBATION EFFECTS

To explore effective perturbation patterns, we begin by considering their relationship to the question.

**Fidelity-Diversity Tradeoff Principle.** With our theory at the ready, we now discuss several aspects of sampling diversity. From the results of Theorem 3.5, we examine the error reduction term $C_N = \Omega\big(\hat{\mu}_1^2 N / (1 + \epsilon)\big)$. For a larger improvement, the prompt perturbations should induce a high first-moment lower bound $\hat{\mu}_1$ while maintaining a low fidelity loss rate $\epsilon$. This suggests a sweet spot where injected prompts are different enough to create exploration (boost $\hat{\mu}_1$), but faithful enough to avoid harming single-attempt quality (control $\epsilon$).

**Observation.** From the above theoretical analysis, we conducted an empirical study. We prompt a model to generate $N$ solution ideas, which are injected into the original prompt as perturbations. We analyze five perturbation styles that are characterized by varying levels of fidelity to the question: Perturbation 1, ideas entirely unrelated to the task, for example generating baking recipe flavor suggestions even when the task is math or code; Perturbation 2, loosely related ideas that differ in topic from the task domain but still connect to learning or knowledge;

Perturbation 3, directly relevant ideas that align with the question itself; Perturbation 4, rephrasings of the original question; and Perturbation 5, verbatim repetition of the original question. The prompts used to obtain these perturbations are provided in Appendix G.1.

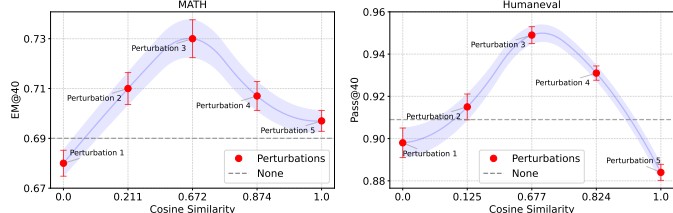

Figure 2: **Effect of perturbation relevance.** Relationship between perturbation-question similarity and task performance. EM rate (math) and Pass rate (code) measured from 40 solutions under five perturbation types (1–5). Results are obtained with `GPT-4o-mini` and are reported as the mean and standard deviation over five independent runs.

We computed the average embedding cosine similarity between these perturbations and the question content using `all-MiniLM-L6-v2` (Reimers & Gurevych, 2019), as well as their EM rate and Pass rate. As shown in Figure 2, we find that EM and Pass rates vary non-monotonically with perturbation relevance. Performance exhibits a rise-then-fall pattern: irrelevant ideas (Perturbation 1) and verbatim repetition (Perturbation 5) fail to improve performance and may even degrade it, while performance increases with higher relevance, peaks with task-aligned ideas (Perturbation 3), and then declines again when similarity becomes excessive. These observations are consistent with our fidelity-diversity trade-off principle. We analyze this in more detail in Appendix B.5.1.

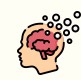 *The relationship between perturbation relevance and scaling inference performance is non-linear. Moderately relevant perturbations contribute positively, whereas overly low or excessively high similarity offers no benefit and may even degrade performance.*

## 4.2 PERTURBATION INSTANTIATIONS

Section 4.1 tells that only meaningful perturbations are effective. Guided by this, we outline two categories of perturbations: task-level and query-level. Task-level perturbations are task-dependent but independent of specific questions, whereas query-level perturbations directly tied to the questions.

**Task-level Perturbations.** These perturbations are independent of specific question content and are sampled from a pool of predefined candidates. (1) **Role** injection samples predefined identity-descriptive sentences (e.g., "mentor", "optimizer", "innovator") into prompts (Shanahan et al., 2023; Kong et al., 2024), steering the model to generate outputs aligned with different personas. (2) **Strategical Instruction** injection introduces stepwise guidance or problem-solving heuristics (Zhou et al., 2023b; Cook et al., 2020; Naik et al., 2023), steering the model toward generating logical and contextually aligned outputs. We refer readers to Appendix G.2 for detailed descriptions.

**Query-level Perturbations.** To obtain more meaningful perturbations, we consider two strategies: (1) **Random Idea Injection** (`RandIdeaInj`), where an LLM (either the target model or another) acts as a thinker to propose task-related ideas which are then injected into the original prompt for perturbation. (2) **Random Query Rephraser** (`RandQReph`) restates the input question (Deng et al., 2023), yielding a modified query $q'_k$ that replaces $q_k$ in $(p, q'_k)$. Rephrasing can also be achieved through back-translation (Beddiar et al., 2021), which produces alternative phrasings while maintaining contextual consistency. Both strategies support three variants: **Single**, where the model itself generates ideas or rephrasings; **Dual**, where a separate model is used; and **Diverse**, where a pool of models each provides varied perturbations in advance, and at each iteration the perturbation is selected from this perturbation set.

## 5 WHEN DIVERSIFIED PERTURBATIONS ARE EFFECTIVE

In this section, we study when diversified exploration remains effective under different conditions. Specifically, we examine its behavior under varying sampling temperatures, different thinker models, different perturbation cardinalities, the presence or absence of Chain-of-Thought (CoT) reasoning, and different choices of verifier. Our empirical evaluation primarily uses the perturbations introduced

in Section 4.2, comparing them against direct sampling (**None**) and the **Jabberwocky**[1] baseline. For each setting, we provide empirical analyses and highlight the corresponding takeaways; due to page limits, the theoretical analyses are deferred to the appendix.

## 5.1 BEHAVIOR UNDER DIFFERENT CONDITIONS

**Condition-I: Varying Sampling Temperatures.** Sampling temperature is a widely used decoding parameter that directly controls the randomness of generation, and prior work has employed temperature scheduling to increase solution diversity (Holtzman et al., 2019; Zhu et al., 2025; Ahmed et al., 2025). We investigate the effectiveness of perturbations across varying sampling temperatures. Specifically, we

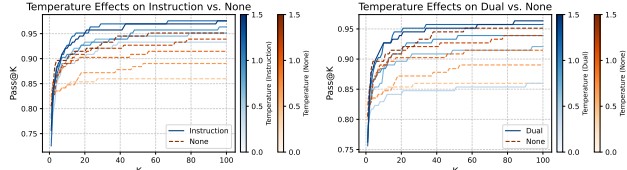

Figure 3: Sweep over a range of increasing temperature settings on Humaneval using `GPT-4o-mini`. Higher temperatures generally improve Pass@k for direct sampling, and diversified sampling provides further gains on top of these temperature-induced improvements at each setting.

evaluate the task-level perturbation **Strategical Instruction** and the **Dual** variant of the query-level perturbation `RandIdeaInj` under varying temperature settings to assess their performance. Figure 3 shows the results on the Humaneval dataset for temperatures $\{0.0, 0.4, 0.8, 1.2, 1.4\}$. Our findings show that perturbations and direct sampling all exhibit improvements at higher temperatures. Instruction attains the highest Pass@100, outperforming the best performance of direct sampling by 2.6%, while Dual's best Pass@100 exceeds the direct sampling maximum by 1.3%.

**Condition-II: Varying Thinker Models.** We analyze the impact of different thinker models in query-level perturbation. Figure 4 shows the performance of the **Dual** variant of the query-level perturbation `RandIdeaInj` across different thinker models, using `GPT-4o-mini` as the policy model. The results indicate that stronger thinker models, such as DeepSeek-V3 (Liu et al., 2024), raise the scaling curve. We suggest that practitioners choose the thinker model according specific use cases and resource constraints.

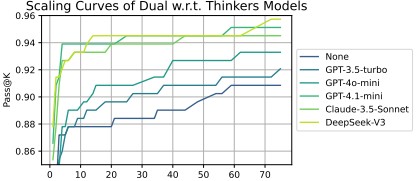

Figure 4: Scaling curves of the Dual strategy across thinker models, with stronger models yielding higher performance.

**Condition-III: Varying Perturbation Cardinality.** We analyze the effect of varying the number of perturbations. Figure 5 shows the Dual strategy on HumanEval with 1, 50, and 100 distinct injection ideas, where `GPT-4o-mini` generates the solutions and `GPT-3.5-turbo` serves as the thinker. We find that increasing the number of injection ideas raises the scaling curve, whereas using only a single injection candidate yields a noticeably lower curve. These observations are in line with our theory: the larger the moment $\hat{\mu}_1$ in Assumption 3.1 are, the more significant the improvement of diversified sampling over regular sampling is. We leave more detailed discussions of this relation to Remark B.6 in Appendix B.

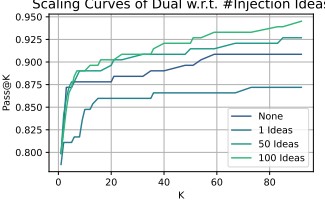 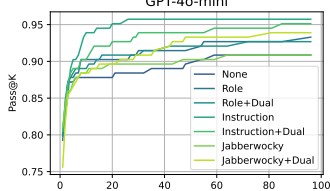 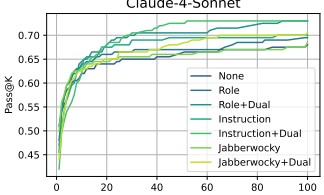

Figure 5: Scaling curves for the Dual strategy as injection cardinality increases.

Figure 6: Perturbations with Chain-of-Thought (CoT). *Left*. HumanEval results using `GPT-4o-mini`. *Right*. APPS results using `Claude-4-Sonnet`.

**Condition-IV: With Chain-of-Thought Reasoning.** Chain-of-Thought (CoT) prompting structures reasoning into explicit intermediate steps, helping LLMs arrive at more accurate answers on complex

---

[1] Jabberwocky is a classic nonsense poem, often regarded as a prototypical example of meaningless text (Carroll, 1978); the details are in Appendix G.2.

reasoning (Wei et al., 2022; Wang et al., 2023b). Building on CoT, we introduce perturbations and conduct analyses with `GPT-4o-mini` and `Claude-4-Sonnet`. We evaluate task-level perturbations, Role and Instruction, along with their combinations with the query-level strategy Dual using the `GPT-3.5-turbo` thinker. These are compared against the random perturbation Jabberwocky and direct sampling without perturbations. All perturbations are applied under the Chain-of-Thought (CoT) setting, with prompt details provided in Appendix G.3. Results in Figure 6 show that task-level and query-level perturbations improve performance under CoT, yielding up to a 4.7% relative gain in Pass@100 on HumanEval with `GPT-4o-mini` and a 7.4% relative gain on APPS with `Claude-4-Sonnet`.

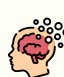

*Diversified perturbations remain effective across a wide range of conditions. Specifically: (i) as the **sampling temperature** increases, scaling curves improve, and diversified perturbations consistently yield additional gains on top of these temperature-induced improvements; (ii) under the **CoT** setting, diversified perturbations further enhance inference performance; and (iii) for query-level perturbations, performance is influenced by both the strength of the **thinker model** and the **perturbation cardinality**: scaling improves as thinker models become stronger and as the number of perturbations increases.*

## 5.2 EFFECT OF VERIFICATION

Although ground-truth reward is commonly used (Zhong et al.; Lewkowycz et al., 2022; Wang et al., 2023c), considering a solution correct if it matches the hidden answer set or passes all hidden tests, outcome reward models (ORMs) offer an alternative by scoring candidate solutions (Zhong et al.; Lightman et al., 2023). However, this approach relies heavily on the quality of the ORM itself. In practice, the ORM's own model performance may introduce biases or errors, and how to train a truly reliable ORM for the verification phase remains an open question that lies outside the scope of our study. To avoid potential interference from external ORMs, one alternative verifier is to use the model itself to evaluate and select the final solution. We evaluate task-level perturbations on MATH and HumanEval using `GPT-4o-mini`. For reasoning and math tasks, the model evaluates and scores each solution, and the top-10 highest-scoring ones can be selected; the instance is marked correct if any of them exactly matches the ground truth (Figure 7(a)). For code generation tasks, the model is prompted to produce 10 unit tests, and a solution is marked correct if it passes at least one of them (Figure 7(b)). The ORM prompt templates are provided in Appendix G.5. The results show that perturbations remain effective when using the LLM-as-a-judge verifier. We also analyze the effectiveness of ORM from our theoretical standpoint: when the ORM has high quality, the ORM pass@$k$ rate is very close to the oracle pass@$k$ rate. See detailed analyses in Appendix B.5.2.

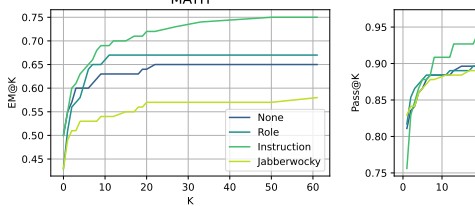
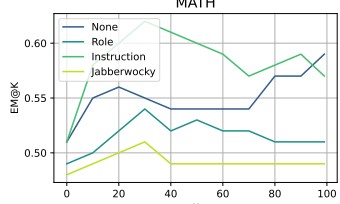

Figure 7: Perturbations are evaluated using the LLM-as-a-Judge verifier. *Left.* Solutions from MATH are rated and selected by the model. *Right.* Solution from Humaneval are evaluated on 10 generated unit tests.

Figure 8: Perturbations are evaluated on MATH, with final answers derived by a majority-voting verifier.

Another verification approach is the majority-voting reward, where the most frequent candidate solution is treated as the final answer. In current verifier-free test-time scaling, there is broad consensus that majority voting is one of the most effective and robust strategies, commonly used in conjunction with repeated sampling (Wang et al., 2025b; Liu et al., 2025; Ghosal et al., 2025; Brown et al., 2024; Wang et al., 2023c; Chen et al., 2025). However, we identify, and analyze theoretically, a failure mode of majority-voting–based verifier-free test-time scaling: unlike pass@$k$, majority voting does not enjoy the same performance boost from repeated sampling and can even degrade

performance in the worst case (see Appendix B.5.3). This can be loosely inferred via the following simple proposition:

**Proposition 5.1.** *Given one-shot correct probability $p(y^*)$ where $y^*$ is the correct answer in response set $Y$, as $N \to \infty$, majority vote accuracy converges to $1$ iff $p(y^*) = \max_{y \in Y} p(y)$, while Best-of-$N$ accuracy converges to $1$ iff $p(y^*) > 0$.*

From here, the key is the underlying assumption of the evaluation protocol: for Best-of-$N$ inference without counting majority, diversity is beneficial because it increases the likelihood that at least one correct solution appears among the $N$ attempts, increasing the chance that $p(y^*) > 0$, hence greater divergence is desirable. Under a majority-voting reward, however, performance improves only when the majority of solution candidates produce the correct solution; mere solution divergence therefore does not guarantee any benefit, and may even reduce $p(y^*)$ to become non-maximal. To empirically validate this failure mode, we conduct experiments on the MATH, where final answers are well-defined and thus naturally suited to majority voting. Figure 8 reports the performance of perturbations under a majority-voting reward on MATH, where we find that perturbations do not yield consistent improvements over direct sampling, thereby providing further empirical support for our theoretical analysis.

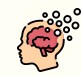

> *Diversified sampling appear effective under the LLM-as-a-Judge setting, yet their benefits may not hold under majority-voting reward as performance gains require convergence toward the correct solution rather than diversity.*

## 6 How Effective Are Diversified Perturbations

In this section, we empirically evaluate how effective, when applied appropriately, diversified sampling is across reasoning, mathematics, and code-generation tasks.

### 6.1 Evaluation Setup

**Datasets.** We evaluate perturbations across six benchmarks spanning reasoning ( **MMLU-Pro** (Wang et al., 2024)), mathematics (**GSM-Hard** (Gao et al., 2023), **MATH** (Hendrycks et al., 2021b)), and code generation (**HumanEval** (Chen et al., 2021), **MBPP** (Austin et al., 2021), **APPS** (Hendrycks et al., 2021a)). Detailed dataset descriptions and setup are provided in Appendix E.1.

**Implementing Details.** For simplicity, we configured the models with a temperature of 0.6 for all datasets. We did not use nucleus sampling across the experiments. Perturbations are evaluated against direct sampling without perturbation, denoted as **None** in all experiments. **Jabberwocky**, a random segment of poetry, is used as a reference perturbation. We run experiments on a server with 4 NVIDIA A100 GPUs, each one with 80GB RAM.

### 6.2 Results of Task-level Perturbations

We evaluate the task-level perturbations across six benchmarks spanning reasoning, mathematics, and code generation, comparing them to direct sampling. All method evaluations are allocated the same search budget of 100 solutions. Figure 9 shows their scaling curves of evaluation on `GPT-4o-mini` (OpenAI, 2023a). We find that injections consistently yield improvements across all tasks, with the instruction strategy performing the best, achieving notable increases of 6.7% EM@100 on MMLU-Pro, 9.6% EM@100 on MATH, and 9.5% Pass@100 on MBPP, over direct sampling.

### 6.3 Results of Query-level Perturbations

**Random Idea Injection.** The evaluation involves a range of `RandIdeaInj` strategies in Section 4.2, including the Single, Dual and Diverse variants, evaluated across the benchmarks MMLU-Pro, MATH, and HumanEval. Figure 10 displays the scaling curves of evaluations conducted with the generative model `GPT-4o-mini`. Evaluations are allocated the same search budget of 100 solutions. `RandIdeaInj` exhibits consistent improvement with idea-injected prompts, achieving a 10.8% increase in reasoning on MMLU-Pro, a 8.2% increase in the mathematics on MATH, and a 4.7%

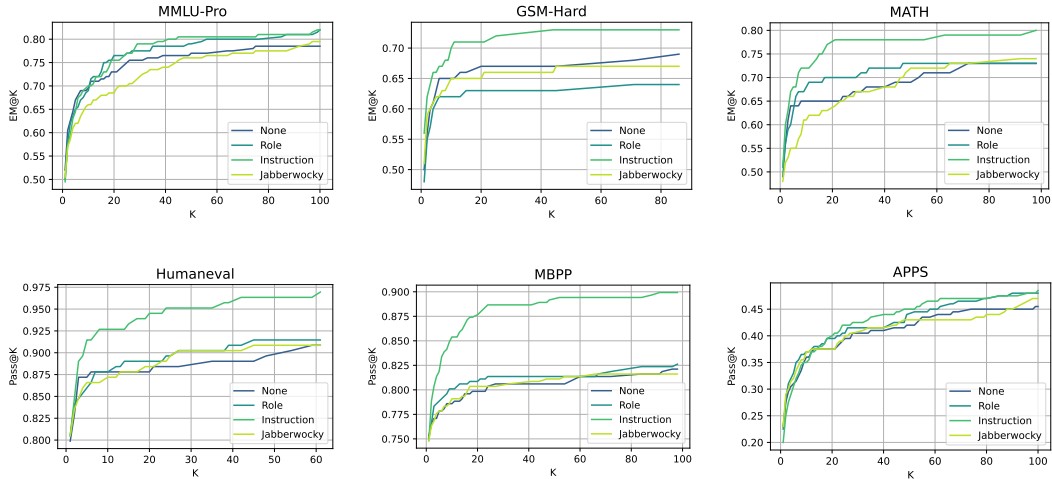

Figure 9: EM@k or Pass@k graphs of task-level perturbations versus direct sampling using `GPT-4o-mini` across six datasets.

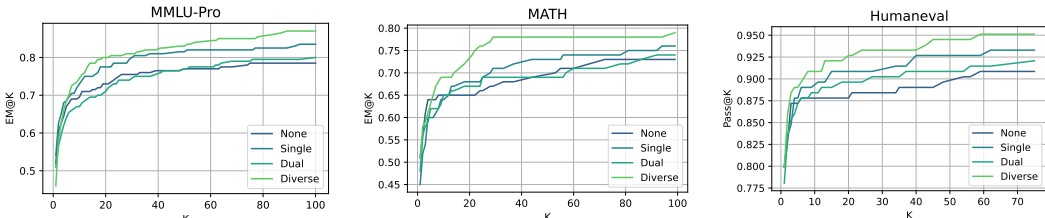

Figure 10: EM@k or Pass@k graphs of Single, Dual and Diverse variants of `RandIdeaInj` versus direct sampling on the MMLU-Pro, MATH and Humaneval benchmarks using `GPT-4o-mini`. In the Dual strategy, `GPT-3.5-turbo` OpenAI (2023a) serves as the thinker. The Diverse method utilizes a set of four models, including `GPT-3.5-turbo`, `GPT-4o-mini`, `Llama-3.1-8B-Instruct` Meta (2024) and `Qwen2.5-7B-Instruct` Yang et al. (2024b).

increase in coding on the Humaneval dataset, over the direct sampling. Similarities of diversified solutions are list in Table 2. We observe that higher solution diversity is often accompanied by better performance, which empirically corroborates our Theorem 3.5 that, under mild assumptions, increased diversity improves Best-of-$N$ performance. Additional results for `RandIdeaInj` and its combination with task-level perturbations across models are provided in Appendix E.2.

**Random Query Rephraser.** Results for `RandQReph`, including both rephrasing and back-translation variants, are provided in Appendix E.3.

## 7 ADDITIONAL RELATED WORK

Scaling inference has explored diverse strategies for enhancing LLM capabilities through adaptive test-time compute allocation (Snell et al., 2024; Brown et al., 2024; Manvi et al., 2024; Guan et al., 2025; Chen et al., 2024b). Typically, LLM inference involves decomposing complex questions into sequential intermediate steps that lead to the final answer, exemplified by chain-of-thought (CoT) prompting (Wei et al., 2022; Sprague et al., 2024; Wang & Zhou, 2024) and its variants (Kojima et al., 2022; Zhou et al., 2023a; Wang et al., 2023c; Li et al., 2023a). With the increasing number of steps in a single chain, these methods often suffer from error propagation and struggle with complex computations (Chen et al., 2023). To address the limitation, CoT (Li et al., 2024) has been improved with search-based methods (Zhang et al., 2024b; Yao et al., 2024b; Luo et al., 2024; Light et al., 2025), such as beam search (Xie et al., 2024b) and Best-of-N (Snell et al., 2024). Subsequently, tree search algorithms including MCTS and A* (Yao et al., 2024b; Luo et al., 2024; Zhang et al., 2024a; Hao et al., 2023; Zhou et al., 2024; Choi et al., 2023; Yao et al., 2024a; Chen et al., 2024c;

Table 2: Effects of query-level perturbation variants (Single, Dual, and Diverse) on solution diversity compared to direct sampling.

| Dataset | Strategy | Pass@100 (EM@100) | tf-idf sim. | BERT sim. | Lev. sim. | Seq. sim. |
|---------|----------|-------------------|-------------|-----------|-----------|-----------|
| MMLU-Pro | None | 0.7850 | 0.6565 | 0.9959 | 0.5350 | 0.6047 |
| | Single | 0.8350 | 0.5770 | 0.9930 | 0.4933 | 0.5473 |
| | Dual | 0.8000 | 0.6517 | 0.9959 | 0.5110 | 0.5961 |
| | Diverse | 0.8700 | 0.5688 | 0.9916 | 0.4838 | 0.5299 |
| MATH | None | 0.7300 | 0.7248 | 0.9971 | 0.5801 | 0.6819 |
| | Single | 0.7600 | 0.7188 | 0.9969 | 0.5771 | 0.6756 |
| | Dual | 0.7400 | 0.7193 | 0.9968 | 0.5782 | 0.6777 |
| | Diverse | 0.7900 | 0.6615 | 0.9960 | 0.5493 | 0.6344 |
| HumanEval | None | 0.9085 | 0.1907 | 0.9996 | 0.1941 | 0.1928 |
| | Single | 0.9329 | 0.1717 | 0.9993 | 0.1822 | 0.1846 |
| | Dual | 0.9207 | 0.1728 | 0.9993 | 0.1820 | 0.1863 |
| | Diverse | 0.9512 | 0.1525 | 0.9991 | 0.1582 | 0.1604 |

Xie et al., 2024a; Zhang et al., 2025) further introduced diversity into inference computation by exploring multiple reasoning paths at different levels. Besides, (Zhou et al., 2025) develops a prefix-tree framework for analyzing and tuning truncation-based decoding, enabling a trade-off between diversity and risk and guiding the selection of sampling strategies and parameters. In parallel, diverse prompting strategies (Li et al., 2023a; Mu et al., 2024; Naik et al., 2023; Hu et al., 2025) have been developed to elicit multiple CoT reasoning trajectories, which ultimately converge to more accurate solutions for complex reasoning tasks. While these methods show that extended inference-time search improves performance, they do not systematically examine the effect of exploration diversity in scaling inference. In this paper, we study why sampling diversity benefits Best-of-$N$, how to promote it through perturbations, when diversified sampling succeeds, and how effective it is across tasks. See Appendix F for extended discussion of literature.

## 8 CONCLUSION

As non-determinism receives growing attention and is increasingly exploited in LLM inference-time scaling, this work provides **a systematic analysis** of the effect of sampling diversity. We offered a theoretical perspective on **why** exploration diversity enhances Best-of-$N$ performance. Building on our main theorem, we derive a diversity–fidelity trade-off principle that guides the design of sampling strategies that introduce diversity while preserving fidelity. This principle is supported by our empirical observation that moderately meaningful perturbations are particularly effective, while overly similar or irrelevant ones yield no performance gains and can even lead to degradation. Following this guidance, we instantiate a set of perturbation styles. We theoretically and empirically analyze **when** diversified exploration remains effective, showing that sampling diversity remain effective across a wide range of conditions, while its benefits still depend on the thinker model's strength, the perturbation cardinality, the policy model's strength, the computational budget, and how many perturbations are generated per query. Our theoretical proposition shows that majority voting constitutes a failure mode in which diversity does not lead to performance gains, a behavior that is also confirmed empirically, suggesting that diversity cannot be applied indiscriminately. We conduct extensive and systematic evaluations of sampling diversity on reasoning, mathematics, and code generation tasks, quantifying the performance gains it can deliver. Finally, we summarize several practical considerations in Appendix H. Our systematic analysis provides a theoretical and empirical foundation for **understanding** the effect of diversity in LLM inference-time scaling.

---

The "amnesia" icon used in the text box is by Vitaliy Gorbachev, from Flaticon.

ETHICS STATEMENT

All authors of this paper confirm that they have read and pledged to uphold the ICLR Code of Ethics. This study focuses on evaluating the effect of diversified sampling in LLM inference, with the goal of advancing the understanding of test-time scaling. Our experiments are conducted solely on publicly available benchmarks for reasoning, mathematics, and code generation, without involving human subjects or sensitive data. We aim to contribute to the responsible development of advanced AI technologies by analyzing techniques that enhance inference efficiency and reliability.

REPRODUCIBILITY STATEMENT

For detailed reproducibility information, including full implementation details, hyperparameters, and evaluation protocols, please refer to the main text and the appendix. All proofs are presented in the main text and appendix with detailed explanations and assumptions. We carefully report implementation details to facilitate verification, with the aim of ensuring that all results can be reliably reproduced and extended by the community. All source code, data, and configuration files will be released to ensure the full reproducibility of our results.

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

# APPENDIX

## A  THE USE OF LLMS

We use LLMs as general-purpose assistants for language editing and refinement of manuscript drafts.

## B  THEORETICAL DETAILS

### B.1  PRELIMINARIES AND NOTATION

In this entire section, we use upper case letters $K$ for the number of attempts in the analysis below and lower case letters $k$ for attempt indices. Inputs are $\boldsymbol{r} = [\boldsymbol{p}, \boldsymbol{q}]$ with $\boldsymbol{r} \sim \mathcal{R} := \{\boldsymbol{p}\} \times \mathcal{Q}$ and support R. Each attempt is configured by a *hybrid* mode $\zeta = (\eta, \xi)$, where

- $\eta \sim \nu_0$ is the *base* decoding randomness/configuration (e.g., seed, sampler, temperature, CoT style);
- $\xi \sim \Pi$ is an *auxiliary* diversity source (to be instantiated later).

Given $(\boldsymbol{r}, \eta, \xi)$, the solver draws $\boldsymbol{s} \sim \text{LLM}(\cdot \mid \boldsymbol{r}, \eta, \xi)$ and the verifier $V$ returns $0/1$. Define the per–mode failure log–probability

$$q(\boldsymbol{r}, \eta, \xi) := \log \mathbb{P}_{\boldsymbol{s} \sim \text{LLM}(\cdot \mid \boldsymbol{r}, \eta, \xi)}\big[V(\boldsymbol{s}) = 0\big]. \tag{7}$$

**Unsolvable sets.**  Let

$$\mathrm{R}_0 := \big\{\boldsymbol{r} \in \mathrm{R} : \mathbb{P}[V(\boldsymbol{s}) = 0 \mid \boldsymbol{r}, \eta, \xi^0] = 1 \text{ for } \eta \sim \nu_0\big\}, \tag{8}$$

$$\mathrm{R}_0^\star := \big\{\boldsymbol{r} \in \mathrm{R} : \mathbb{P}[V(\boldsymbol{s}) = 0 \mid \boldsymbol{r}, \eta, \xi] = 1 \text{ for all } \eta, \xi\big\}. \tag{9}$$

Here $\xi^0$ is a fixed auxiliary reference used by the baseline policy; clearly $\mathrm{R}_0^\star \subseteq \mathrm{R}_0$. We denote by $\mu_{\mathcal{R}}$ the probability measure induced by $\mathcal{R}$ on R.

**Best-of-$K$ failure.**  For a policy that samples $K$ i.i.d. modes $(\eta_k, \xi_k)$ and produces $\boldsymbol{s}_k \sim \text{LLM}(\cdot \mid \boldsymbol{r}, \eta_k, \xi_k)$, the failure probability is

$$P^K := \mathbb{P}\Big[V(\boldsymbol{s}_k) = 0, \ \forall k \in [K] \ \Big| \ \boldsymbol{r} \sim \mathcal{R}, \ (\eta_k, \xi_k) \text{ i.i.d.}\Big]. \tag{10}$$

We write $P_{\text{reg}}^K$ for the *baseline* where $(\eta_k, \xi_k) = (\eta_k, \xi^0)$ with $\eta_k \overset{\text{i.i.d.}}{\sim} \nu_0$, and $P_{\text{div}}^K$ for the *hybrid diversified* case $(\eta_k, \xi_k) \overset{\text{i.i.d.}}{\sim} \nu_0 \times \Pi$. Their limits as $K \to \infty$ are $P_{\text{reg}}^{\text{inf}} = \mu_{\mathcal{R}}(\mathrm{R}_0)$ and $P_{\text{div}}^{\text{inf}} = \mu_{\mathcal{R}}(\mathrm{R}_0^\star)$.

### B.2  HYPOTHESIS 1: DISPERSION UNDER AUXILIARY DIVERSITY

We formalize the dispersion contributed by the auxiliary source while keeping the base randomness.

**Assumption B.1** (Dispersion first moment). There exists $\hat{\mu}_1 > 0$ such that for all $\boldsymbol{r} \notin \mathrm{R}_0^\star$,

$$M_1(\boldsymbol{r}; \nu_0, \Pi) := \mathbb{E}_{\eta \sim \nu_0} \mathbb{E}_{\xi \sim \Pi} \Big| q(\boldsymbol{r}, \eta, \xi) - \mathbb{E}_{\xi \sim \Pi} q(\boldsymbol{r}, \eta, \xi) \Big| \geq \hat{\mu}_1. \tag{11}$$

Optionally, one may also posit a (weaker) second-moment lower bound $\mathbb{E}_{\eta, \xi}\big(q - \mathbb{E}_\xi q\big)^2 \geq \hat{\mu}_2 > 0$; our bounds only require the first moment.

*Remark* B.2. Intuitively, unless $\boldsymbol{r}$ is genuinely unsolvable for *all* $(\eta, \xi)$, varying the auxiliary $\xi$ changes the failure likelihood while the base randomness $\eta$ remains at its usual variability.

**Variance decomposition (used implicitly).**  By the law of total variance applied to $q(\boldsymbol{r}, \eta, \xi)$ with independent $\eta \sim \nu_0$, $\xi \sim \Pi$,

$$\text{Var}_{\eta, \xi}\big(q(\boldsymbol{r}, \eta, \xi)\big) = \mathbb{E}_\eta\Big[\text{Var}_\xi\big(q(\boldsymbol{r}, \eta, \xi)\big)\Big] + \text{Var}_\eta\Big(\mathbb{E}_\xi[q(\boldsymbol{r}, \eta, \xi)]\Big) \geq \mathbb{E}_\eta\Big[\text{Var}_\xi\big(q(\boldsymbol{r}, \eta, \xi)\big)\Big]. \tag{12}$$

Assumption B.1 ensures the *absolute* first central moment over $\xi$ is uniformly bounded away from zero after averaging over $\eta$, hence dispersion is nontrivial.

### B.3 HYPOTHESIS 2: FIDELITY WITH FIXED AUXILIARY SETTING

We compare policies under the *same* base randomness, and with auxiliary fixed on both sides.

**Assumption B.3** (Localized fidelity). There exist $\epsilon \in [0, 1)$ such that for all $K \geq 1$ and auxiliary settings $\xi^\star, \xi^0$,

$$\frac{\mathbb{E}_{\boldsymbol{r} \sim \mathcal{R}} \mathbb{E}_{\eta \sim \nu_0} \Big[ \exp\{K\, q(\boldsymbol{r}, \eta, \xi^\star)\} \Big]}{\mathbb{E}_{\boldsymbol{r} \sim \mathcal{R}} \mathbb{E}_{\eta \sim \nu_0} \Big[ \exp\{K\, q(\boldsymbol{r}, \eta, \xi^0)\} \Big]} \leq 1 + \epsilon. \tag{13}$$

*Remark* B.4. The auxiliary choice is *fixed* on both sides so that both policies share the same base variability ($\eta \sim \nu_0$). This guards against conflating auxiliary gains with changes in base decoding. To make this assumption more realistic, one may assume this only holds for a majority of auxiliary settings $\xi^*$, and the proof can be easily revised to account for this softness in assumption. For simplicity, however, we choose to not incorporate this fine-grained approach.

### B.4 MAIN THEOREM AND PROOF

We restate the main comparison bound in the hybrid framework and prove it. Same as in the main text, we write Best-of-$N$ failure probabilities as

$$P_{\text{div}}^N := \mathbb{P}\Big[V(\boldsymbol{s}_k) = 0, \ \forall k \in [N] \ \Big| \ \boldsymbol{s}_k \sim \text{LLM}(\cdot \mid \boldsymbol{r}, \zeta_k), \ \zeta_k \overset{\text{i.i.d.}}{\sim} \nu, \ \boldsymbol{r} \sim \mathcal{R}\Big], \tag{14}$$

$$P_{\text{reg}}^N := \mathbb{P}\Big[V(\boldsymbol{s}_k) = 0, \ \forall k \in [N] \ \Big| \ \boldsymbol{s}_k \sim \text{LLM}(\cdot \mid \boldsymbol{r}, \zeta), \ \zeta \sim \nu_0 \times \{\zeta_0\}, \ \boldsymbol{r} \sim \mathcal{R}\Big], \tag{15}$$

and assume they converge to limits $P_{\text{div}}^{\text{inf}}$ and $P_{\text{reg}}^{\text{inf}}$ as $N \to \infty$.

**Theorem B.5** (Hybrid diversity improves Best-of-$K$). *Under Assumptions B.1 and B.3, there exists a sequence $C_K = \Omega\big(\hat{\mu}_1^2 K / (1 + \epsilon)\big)$, positive and increasing in $K$, such that*

$$P_{\text{div}}^K \leq \frac{P_{\text{reg}}^K - P_{\text{reg}}^{\text{inf}}}{1 + C_K} + P_{\text{div}}^{\text{inf}}, \qquad \text{with} \quad P_{\text{div}}^{\text{inf}} \leq P_{\text{reg}}^{\text{inf}}. \tag{16}$$

*Proof.* Write $q_k := q(\boldsymbol{r}, \eta_k, \xi_k)$, where $(\eta_k, \xi_k) \overset{\text{i.i.d.}}{\sim} \nu_0 \times \Pi$ for the diversified policy. By independence of attempts,

$$P_{\text{div}}^K = \mathbb{E}_{\boldsymbol{r} \sim \mathcal{R}} \mathbb{E}_{(\eta_k, \xi_k)} \Big[ \exp\Big\{ \sum_{k=1}^K q_k \Big\} \Big]. \tag{17}$$

Introduce the sample mean $\bar{q} := \frac{1}{K} \sum_{k=1}^K q_k$ and the mixed mean $\bar{q}_\Pi(\boldsymbol{r}) := \mathbb{E}_{\eta, \xi}\, q(\boldsymbol{r}, \eta, \xi)$ (distinct from $\bar{q}$). For $g(x) = e^x - x - 1 \geq \min\{0.25x^2, 0.5|x|\}$,

$$\frac{\frac{1}{K} \sum_{k=1}^K \exp\{K q_k\}}{\exp\{\sum_{k=1}^K q_k\}} = \frac{1}{K} \sum_{k=1}^K \exp\{K(q_k - \bar{q})\} = \frac{1}{K} \sum_{k=1}^K \big(1 + K(q_k - \bar{q}) + g(K(q_k - \bar{q}))\big). \tag{18}$$

Hence

$$\frac{\frac{1}{K} \sum_{k=1}^K \exp\{K q_k\}}{\exp\{\sum_{k=1}^K q_k\}} \geq 1 + \min\Big\{0.5 \sum_{k=1}^K |q_k - \bar{q}|, \ 0.25\, K \sum_{k=1}^K (q_k - \bar{q})^2\Big\}. \tag{19}$$

By Hoeffding (or Bernstein) and Assumption B.1, with probability at least $1 - \delta$ (over draws of $(\eta_k, \xi_k)$) we have, uniformly for $\boldsymbol{r} \notin \text{R}_0^\star$,

$$\sum_{k=1}^K |q_k - \bar{q}| \geq \hat{\mu}_1 K - C\sqrt{K \log(1/\delta)}, \tag{20}$$

$$\sum_{k=1}^K (q_k - \bar{q})^2 \geq \frac{1}{K}\Big(\sum_{k=1}^K |q_k - \bar{q}|\Big)^2 \geq \hat{\mu}_1^2 K - C'\sqrt{K \log(1/\delta)}. \tag{21}$$

Plugging equation 20–equation 21 into equation 19, and absorbing deviations into constants $C_1, C_2$ independent of $r$, yields

$$\frac{\frac{1}{K}\sum_{k=1}^{K}\exp\{Kq_k\}}{\exp\{\sum_{k=1}^{K}q_k\}} \geq 1 + C_1 K - C_2 \sqrt{K}. \tag{22}$$

Hence, conditioning on $r \notin \mathrm{R}_0^\star$ and taking expectations,

$$\mathbb{E}\Big[\exp\Big\{\sum_{k=1}^{K}q_k\Big\}\,\Big|\,r\Big] \leq \frac{1}{1+C_1 K - C_2\sqrt{K}} \cdot \frac{1}{K}\sum_{k=1}^{K}\mathbb{E}\big[\exp\{Kq_k\}\,\big|\,r\big]. \tag{23}$$

Now apply Assumption B.3 with $(\eta_k, \xi_k) \sim \nu_0 \times \Pi$ versus baseline $(\eta_k, \xi^0)$, and average over $r \sim \mathcal{R}$:

$$\frac{1}{K}\sum_{k=1}^{K}\mathbb{E}_r\mathbb{E}_{\eta,\xi}\big[\exp\{Kq(r,\eta,\xi)\}\big] \leq (1+\epsilon)\,\mathbb{E}_r\mathbb{E}_\eta\big[\exp\{Kq(r,\eta,\xi^0)\}\big]. \tag{24}$$

Combining equation 17, equation 23, equation 24, and then splitting the expectation over $\mathrm{R}$ into $\mathrm{R}\setminus\mathrm{R}_0^\star$ and $\mathrm{R}_0^\star$ (where $\exp\{\sum q_k\} \equiv 1$), we obtain

$$P_{\text{div}}^K - P_{\text{div}}^{\inf} \leq \frac{1+\epsilon}{1+C_1 K - C_2\sqrt{K}}\Big(P_{\text{reg}}^K - P_{\text{reg}}^{\inf}\Big). \tag{25}$$

Letting $C_K := (C_1 K - C_2\sqrt{K} - \epsilon)/(1+\epsilon)$, which is $\Omega(\hat{\mu}_1^2 K/(1+\epsilon))$ by construction of $C_1$ through equation 20–equation 21, yields equation 16. Finally, $P_{\text{div}}^{\inf} = \mu_\mathcal{R}(\mathrm{R}_0^\star) \leq \mu_\mathcal{R}(\mathrm{R}_0) = P_{\text{reg}}^{\inf}$ by equation 8. □

*Remark* B.6 (Moment effect and asymptote). The leading term of $C_K$ scales as $\Theta\big(\hat{\mu}_1^2 K/(1+\epsilon)\big)$: richer (but faithful) auxiliary diversity increases the dispersion (larger $\hat{\mu}_1$), which steepens the Best-of-$K$ contraction; simultaneously, the unsolvable set shrinks from $\mathrm{R}_0$ to $\mathrm{R}_0^\star$ so the limiting failure $P_{\text{div}}^{\inf}$ decreases.

## B.5 Additional Theoretical Discussions

With our theory at the ready, we now discuss several aspects of diversified sampling to support some of our empirical findings.

### B.5.1 Perturbation–Question Similarity and the Diversity–Fidelity Tradeoff

Empirically (Fig. 2), EM/Pass improve as perturbations become more relevant to the question, peak around task-aligned ideas, then drop when relevance becomes excessive (rephrasings/verbatim). This matches our theory through the two quantities already in the main text:

**(A) Dispersion $M_1$ shrinks when relevance is *too high*.** When perturbations are near rephrasings or verbatim repeats, they induce very similar solver behavior across attempts, so the failure log-probability barely moves. Hence the first-moment $M_1$ is small, the convergence factor

$$C_N \propto \frac{M_1^2 N}{1+\epsilon}$$

is small, and Best-of-$N$ gains vanish. This explains the decline from Perturbation 4 (rephrasing) to Perturbation 5 (verbatim).

**(B) Fidelity $\epsilon$ worsens when relevance is *too low*.** When perturbations are off-topic or loosely related, single-attempt quality degrades. In our bound this appears as a larger $\epsilon$, which *divides* the gain: even if diversity increases, the $(1+\epsilon)$ penalty suppresses $C_N$ and reduces overall improvement. This explains the weak performance of Perturbation 1 (irrelevant) and the partial recovery at Perturbation 2 (loosely related).

**(C) The sweet spot is *task-aligned* ideas.** At intermediate relevance, perturbations are different enough to move the model into alternative solution modes (so $M_1$ is substantial) while still faithful to the task (so $\epsilon$ remains modest). This maximizes the effective gain $M_1^2/(1+\epsilon)$, producing the peak at Perturbation 3.

**Design takeaway.** Concentrate perturbations in a *moderate relevance band*: different enough to create exploration (boost $M_1$), but faithful enough to avoid harming single-attempt quality (control $\epsilon$). This is exactly where our experiments see the maximum Best-of-$N$ improvement.

### B.5.2 OUTCOME REWARD MODELS (ORM) AND WHY TOP-$k$ CAPTURES THE BEST

Let a single instance produce $N$ candidate solutions $S = \{s_1, \ldots, s_N\}$. The oracle (ground truth) verifier $V$ labels a solution correct if it matches the hidden answer (or passes hidden tests). An *outcome reward model* (ORM) assigns a real score $R(s)$ to each $s$, used to rank candidates; the top-$k$ by $R$ are selected.

We analyze the event that the *best* correct solution $s^\star$ (if one exists) appears in the top-$k$ by $R$. Write a simple low-error model for the ORM:

$$R(s) = \mu(s) + \varepsilon_s, \qquad \varepsilon_s \text{ i.i.d. sub-Gaussian with variance proxy } \sigma^2, \tag{26}$$

and assume a *margin* $\gamma > 0$ between $s^\star$ and every incorrect solution $\tilde{s}$ in the latent score:

$$\mu(s^\star) \geq \max_{\tilde{s}:\, V(\tilde{s})=0} \mu(\tilde{s}) + \gamma. \tag{27}$$

**Theorem B.7** (ORM top-$k$ recall under a margin). *Under the model above, for any incorrect $\tilde{s}$,*

$$\Pr\big[R(\tilde{s}) \geq R(s^\star)\big] \leq p_\gamma := \exp\Big(-\tfrac{\gamma^2}{4\sigma^2}\Big). \tag{28}$$

*Let $X$ be the number of incorrect solutions that outrank $s^\star$. Then $\mathbb{E}[X] \leq (N-1)\,p_\gamma$ and, by Markov,*

$$\Pr\big[s^\star \notin \text{Top-}k\big] = \Pr[X \geq k] \leq \frac{(N-1)\,p_\gamma}{k}. \tag{29}$$

*In particular, with $k = 10$,*

$$\Pr\big[s^\star \in \text{Top-10}\big] \geq 1 - \frac{(N-1)}{10}\exp\Big(-\tfrac{\gamma^2}{4\sigma^2}\Big). \tag{30}$$

*Proof of Theorem B.7.* Let $s^\star$ be a correct solution and suppose the latent scores satisfy the margin

$$\mu(s^\star) \geq \max_{\tilde{s}:\, V(\tilde{s})=0} \mu(\tilde{s}) + \gamma, \tag{31}$$

for some $\gamma > 0$. For any incorrect $\tilde{s}$, consider the difference

$$R(\tilde{s}) - R(s^\star) = \big(\mu(\tilde{s}) - \mu(s^\star)\big) + \big(\varepsilon_{\tilde{s}} - \varepsilon_{s^\star}\big) \leq -\gamma + Z_{\tilde{s}}, \tag{32}$$

where $Z_{\tilde{s}} := \varepsilon_{\tilde{s}} - \varepsilon_{s^\star}$ is sub-Gaussian with variance proxy $2\sigma^2$ (since the $\varepsilon$'s are i.i.d. sub-Gaussian with proxy $\sigma^2$).

By the standard sub-Gaussian tail bound, for any $t > 0$,

$$\Pr\big[Z_{\tilde{s}} \geq t\big] \leq \exp\Big(-\tfrac{t^2}{4\sigma^2}\Big). \tag{33}$$

Setting $t = \gamma$ yields

$$\Pr\big[R(\tilde{s}) \geq R(s^\star)\big] \leq \exp\Big(-\tfrac{\gamma^2}{4\sigma^2}\Big) := p_\gamma. \tag{34}$$

Let $I_{\tilde{s}} := \mathbb{1}\{R(\tilde{s}) \geq R(s^\star)\}$ and $X := \sum_{\tilde{s}:\, V(\tilde{s})=0} I_{\tilde{s}}$ be the number of incorrect solutions that outrank $s^\star$. By linearity of expectation and equation 34,

$$\mathbb{E}[X] = \sum_{\tilde{s}:\, V(\tilde{s})=0} \mathbb{E}[I_{\tilde{s}}] \leq (N-1)\,p_\gamma, \tag{35}$$

where we crudely upper bound the number of competitors by $N-1$. By Markov's inequality,

$$\Pr\big[X \geq k\big] \leq \frac{\mathbb{E}[X]}{k} \leq \frac{(N-1)\,p_\gamma}{k}. \tag{36}$$

The event $\{s^\star \notin \text{Top-}k\}$ implies that at least $k$ candidates outrank $s^\star$, hence

$$\Pr\big[s^\star \notin \text{Top-}k\big] \leq \Pr[X \geq k] \leq \frac{(N-1)\,p_\gamma}{k}, \tag{37}$$

which proves the stated bound. For $k = 10$ this gives

$$\Pr\big[s^\star \in \text{Top-10}\big] \geq 1 - \frac{(N-1)}{10}\exp\Big(-\tfrac{\gamma^2}{4\sigma^2}\Big). \tag{38}$$

**Multiple correct solutions.** If there are several correct solutions $\{s_j^\star\}_{j=1}^m$ each separated by the same (or larger) margin $\gamma$ from *all* incorrect solutions, define $X_j$ as the number of incorrect solutions outranking $s_j^\star$. Then by the same argument,

$$\Pr\big[\forall j,\ s_j^\star \notin \text{Top-}k\big] \ \leq\ \Pr\big[\min_j X_j \geq k\big] \ \leq\ \min_j \Pr[X_j \geq k] \ \leq\ \frac{(N-1)\,p_\gamma}{k}, \qquad (39)$$

so the probability that *none* of the correct solutions appears in top-$k$ is no larger than in the single-best case. Consequently, the ORM pass@$k$ differs from oracle pass@$k$ only when all correct solutions are excluded from the ORM's top-$k$, an event controlled by the bound above. $\qquad\square$

**Implications.** If the ORM is *high quality* (small $\sigma$) and the correct solution is reasonably separated (moderate $\gamma$), the chance that top-$k$ by $R$ misses $s^\star$ decays rapidly—in particular, top-10 typically contains the best solution. Consequently, the ORM pass@$k$ closely tracks the oracle pass@$k$ (they differ only when the oracle-best is excluded from the ORM's top-$k$). With multiple correct solutions, the probability that *none* appear in top-$k$ is even smaller (union bound), further tightening alignment between ORM and oracle pass@$k$.

### B.5.3 WHY MAJORITY VOTING DOES NOT MIRROR BEST-OF-$N$ GAINS

**Set-up.** Fix an instance with a discrete answer space $\mathcal{Y}$ (e.g., final numbers in math). Under a sampling policy $\pi$ (e.g., baseline or a diversified hybrid), let

$$p_\pi(y) \ :=\ \Pr_{\boldsymbol{s}\sim\pi}[\text{final answer of } \boldsymbol{s} = y], \qquad y \in \mathcal{Y}, \qquad (40)$$

and denote the ground-truth answer by $y^\star$. Majority voting over $N$ i.i.d. samples chooses the label with the largest empirical frequency.

**Asymptotics of majority vote.** By the law of large numbers, empirical frequencies converge to $\{p_\pi(y)\}_y$. Hence the majority-vote output converges almost surely to

$$\arg\max_{y\in\mathcal{Y}} p_\pi(y). \qquad (41)$$

Therefore, the *asymptotic* majority-vote accuracy equals

$$\lim_{N\to\infty} \Pr[\text{MV}_N = y^\star] \ =\ \mathbf{1}\left\{ p_\pi(y^\star) \ =\ \max_y p_\pi(y) \right\}, \qquad (42)$$

i.e., it is $1$ if and only if the correct answer is already the *most probable* single-sample outcome under $\pi$, and $0$ otherwise.

**Contrast with Best-of-$N$.** Best-of-$N$ success is

$$\Pr[\text{BoN}_N \text{ hits } y^\star] \ =\ 1 - (1 - p_\pi(y^\star))^N, \qquad (43)$$

which increases *monotonically* to $1$ whenever $p_\pi(y^\star) > 0$. Thus, BoN rewards *diversity of attempts* (any nonzero mass on $y^\star$ helps), whereas majority vote rewards *mass concentration* (the correct label must be the single most likely).

**Implication for diversified sampling.** Diversification typically *spreads* probability mass across multiple solution modes. Unless it also raises $p_\pi(y^\star)$ above all competitors, majority voting has no systematic reason to improve—and can degrade if the spread lowers the rank of $y^\star$. This explains the inconsistent gains in Fig. 8.

**A simple bound.** Let $X_y$ be the count of label $y$ among $N$ i.i.d. draws. Then

$$\Pr[\text{MV}_N = y^\star] \ =\ \Pr\big[X_{y^\star} \ \geq\ \max_{y\neq y^\star} X_y\big]. \qquad (44)$$

When $p_\pi(y^\star) \leq \max_{y\neq y^\star} p_\pi(y)$, concentration of measure implies $\Pr[\text{MV}_N = y^\star] \to 0$ as $N \to \infty$; when $p_\pi(y^\star)$ is uniquely maximal, the probability $\to 1$. In contrast, $\Pr[\text{BoN}_N \text{ hits } y^\star] \to 1$ whenever $p_\pi(y^\star) > 0$.

**Relation to fidelity (Hypothesis 3.3).** For fixed auxiliary settings, Hypothesis 3.3 controls the *single-attempt* failure via a factor $(1 + \epsilon)$ across policies. Majority voting asymptotically reduces to a *best@1* decision (choosing the MAP label), so it inherits this limitation: switching to a diversified auxiliary setting cannot guarantee improvement and may even inflate error by up to a $(1 + \epsilon)$ factor relative to baseline, whereas Best-of-$N$ benefits strictly from added dispersion.

## C  DETAILS OF METRICS

For each of our metrics, the solver is allowed $k$ submissions for each, denoted by $[\boldsymbol{s}]_k \sim \text{LLM}(\cdot|\boldsymbol{r}, k)$ given input $\boldsymbol{r}$. We consider testing the model on a set of tasks consisting of prompts and questions $\mathbf{X} = \{\boldsymbol{r} = [\boldsymbol{p}, \boldsymbol{q}]\}$.

**EM@k Rate**. For reasoning and math tasks, if at least one submission $s' \in [\boldsymbol{s}]_k$ matches the ground truth, the task is considered solved. The EM@k rate is defined as the proportion of tasks solved as

$$\textbf{EM@k} = \frac{1}{|\mathcal{X}|} \sum_{\boldsymbol{r} \in \mathbf{X}} \mathbb{1}\big(\exists \boldsymbol{s} \in [\boldsymbol{s}]_k, \text{s.t.}, \boldsymbol{s} = \boldsymbol{H}\big|[\boldsymbol{s}]_k \sim \text{LLM}(\cdot|\boldsymbol{r}, k)\big),$$

where $\mathbb{1}(\cdot)$ is the indicator function and $\boldsymbol{H}$ is the ground truth.

**Pass@k Rate**. For code generation tasks, if at least one submission $s' \in [\boldsymbol{s}]_k$ passes all hidden tests $\boldsymbol{H}_c$, the task is considered solved. The Pass@k rate is defined as

$$\textbf{Pass@k} = \frac{1}{|\mathcal{X}|} \sum_{\boldsymbol{r} \in \mathbf{X}} \mathbb{1}\big(\exists \boldsymbol{s}' \in [\boldsymbol{s}]_k, \text{s.t.}, \boldsymbol{s}' \text{ passes all } \boldsymbol{H}_c\big|[\boldsymbol{s}]_k \sim \text{LLM}(\cdot|\boldsymbol{r}, k)\big).$$

**TF-IDF Similarity** measures the importance of terms in a document relative to a collection of documents, which computes the average cosine similarity between TF-IDF representations of solution pairs:

$$\textbf{tf-idf sim.} = \frac{1}{|\mathcal{X}|} \sum_{\boldsymbol{x} \in \mathcal{X}} \frac{1}{k(k-1)} \sum_{\substack{\boldsymbol{s}, \boldsymbol{s}' \in [\boldsymbol{s}]_k \\ \boldsymbol{s} \neq \boldsymbol{s}'}} \frac{\text{tf-idf}(\boldsymbol{s}) \cdot \text{tf-idf}(\boldsymbol{s}')}{\|\text{tf-idf}(\boldsymbol{s})\| \, \|\text{tf-idf}(\boldsymbol{s}')\|}.$$

**BERT Cosine Similarity** is an average cosine score between the embeddings of candidate solution pairs, where embeddings are performed using CodeBERT (Feng et al., 2020), a pre-trained model for understanding code semantically:

$$\textbf{BERT sim.} = \frac{1}{|\mathcal{X}|} \sum_{\boldsymbol{x} \in \mathcal{X}} \frac{1}{k(k-1)} \sum_{\substack{\boldsymbol{s}, \boldsymbol{s}' \in [\boldsymbol{s}]_k \\ \boldsymbol{s} \neq \boldsymbol{s}'}} \frac{\text{CodeBERT}(\boldsymbol{s}) \cdot \text{CodeBERT}(\boldsymbol{s}')}{\|\text{CodeBERT}(\boldsymbol{s})\| \, \|\text{CodeBERT}(\boldsymbol{s}')\|}.$$

**Levenshtein Similarity** is based on the Levenshtein distance, which measures the minimum number of single-character edits (insertions, deletions, or substitutions) required to transform one string into another:

$$\textbf{lev. sim.} = \frac{1}{|\mathcal{X}|} \sum_{\boldsymbol{x} \in \mathcal{X}} \frac{1}{k(k-1)} \sum_{\substack{\boldsymbol{s}, \boldsymbol{s}' \in [\boldsymbol{s}]_k \\ \boldsymbol{s} \neq \boldsymbol{s}'}} \frac{\text{LevenshteinDistance}(\boldsymbol{s}, \boldsymbol{s}')}{\max(|\boldsymbol{s}|, |\boldsymbol{s}'|)}.$$

**Token Sequence Similarity** measures the overlap between two sequences of tokens (e.g., programming language tokens), denoted by $T(\boldsymbol{s})$ for output $\boldsymbol{s}$:

$$\textbf{seq. sim.} = \frac{1}{|\mathcal{X}|} \sum_{\boldsymbol{x} \in \mathcal{X}} \frac{1}{k(k-1)} \sum_{\substack{\boldsymbol{s}, \boldsymbol{s}' \in [\boldsymbol{s}]_k \\ \boldsymbol{s} \neq \boldsymbol{s}'}} \frac{|T(\boldsymbol{s}) \cap T(\boldsymbol{s}')|}{|T(\boldsymbol{s}) \cup T(\boldsymbol{s}')|}.$$

## D  ALGORITHMS OF QUERY-LEVEL PERTURBATIONS

Algorithm 1 describes query-level perturbations in their Single and Dual variants. The procedure begins by repeatedly querying a *thinker*, where each query generates a set of $m$ solution ideas for a given question, until a total of $N$ ideas is collected. In the prompt injection step, each idea is incorporated into the original prompt together with the question to construct a perturbed prompt.

---

**Algorithm 1** Query-level Perturbations (Single & Dual).

---

1: **Input:** The sampling LLM, a prompt $p$, question $q$, verifier $V$, repeated sampling times $N$, query size $m$, and the thinker model $t$
2: $\mathbf{S} \leftarrow \varnothing$ ▷ Initialize $\mathbf{S}$ as the set of solutions
3: $r = [p, q]$ ▷ $r$ is the combined input
4: ▷ **Step 0: Generate perturbations from thinker $t$.**
5: $\mathcal{I} \leftarrow \varnothing$ ▷ Initialize $\mathcal{I}$ as the set of solution ideas
6: **while** $|\mathcal{I}| < N$ **do**
7: $\quad \mathcal{M} \leftarrow t(r)$ ▷ Query the thinker $t$ to obtain a set $\mathcal{M}$ of $m$ solution ideas
8: $\quad \mathcal{I} \leftarrow \mathcal{I} \cup \mathcal{M}$
9: **end while**
10: $\mathcal{I} \leftarrow$ first $N$ elements of $\mathcal{I}$
11: **for** $k \leftarrow 1$ to $N$ **do**
12: $\quad$ ▷ **Step 1: Prompt Injection.**
13: $\quad r_k \leftarrow r \oplus i_k$ ▷ Form perturbed prompt $r_k$ by combining $r$ with solution idea $i_k$
14: $\quad$ ▷ **Step 2: Repeated Sampling.**
15: $\quad s_k \leftarrow \text{LLM}(\cdot | r_k)$ ▷ Obtain solution $s_k$
16: $\quad \mathcal{S} \leftarrow \mathcal{S} \cup \{s_k\}$
17: **end for**
18: ▷ **Step 3: Verification.**
19: Sample $s^* \sim \{\, s_k \mid V(s_k) = 1, s_k \in \mathbf{S} \,\}$
20: **Return:** The best answer $s^*$

---

Algorithm 2 presents the Diverse variant. In this perturbation style, we cyclically traverse the thinker pool, using each model in turn to generate a set of solution ideas for the given question, until a total of $N$ ideas is collected. In each iteration of repeated sampling, a perturbation is randomly drawn from the solution idea set and combined with the original prompt and question to construct a perturbed prompt.

---

**Algorithm 2** Query-level Perturbations (Diverse)

---

1: **Input:** The sampling LLM, a prompt $p$, question $q$, verifier $V$, repeated sampling times $N$, query size $m$, and a set of thinkers $\mathcal{T}$
2: $\mathbf{S} \leftarrow \varnothing$ ▷ Initialize $\mathbf{S}$ as the set of solutions
3: $r = [p, q]$ ▷ $r$ is the combined input
4: ▷ **Step 0: Generate perturbations from thinkers.**
5: $\mathcal{I} \leftarrow \varnothing$ ▷ Initialize $\mathcal{I}$ as the set of solution ideas
6: **while** $|\mathcal{I}| < N$ **do**
7: $\quad$ **for** $t \in \mathcal{T}$ **do**
8: $\quad\quad \mathcal{M} \leftarrow t(r)$ ▷ Query thinker $t$ to obtain a set $\mathcal{M}$ of $m$ solution ideas
9: $\quad\quad \mathcal{I} \leftarrow \mathcal{I} \cup \mathcal{M}$
10: $\quad\quad$ **if** $|\mathcal{I}| \geq N$ **then**
11: $\quad\quad\quad$ **break** ▷ Enough ideas collected
12: $\quad\quad$ **end if**
13: $\quad$ **end for**
14: **end while**
15: $\mathcal{I} \leftarrow$ first $N$ elements of $\mathcal{I}$
16: **for** $k \leftarrow 1$ to $N$ **do**
17: $\quad$ ▷ **Step 1: Prompt Injection.**
18: $\quad$ Sample $i_k \sim \mathcal{I}$
19: $\quad r_k \leftarrow r \oplus i_k$ ▷ Form perturbed prompt $r_k$ by combining $r$ with solution idea $i_k$
20: $\quad$ ▷ **Step 2: Repeated Sampling.**
21: $\quad s_k \leftarrow \text{LLM}(\cdot | r_k)$ ▷ Obtain solution $s_k$
22: $\quad \mathcal{S} \leftarrow \mathcal{S} \cup \{s_k\}$
23: **end for**
24: ▷ **Step 3: Verification.**
25: Sample $s^* \sim \{\, s_k \mid V(s_k) = 1, s_k \in \mathbf{S} \,\}$
26: **Return:** The best answer $s^*$

---

**Complexity Analysis.** If we treat each call to the thinker, the sampling LLM, and the verifier as having constant cost, both Algorithm 1 and Algorithm 2 have overall time complexity linear in the number of repeated samples $N$. In Algorithm 1, the thinker is queried $O(N/m)$ times to collect $N$ ideas, and the LLM and verifier are each invoked $O(N)$ times. In Algorithm 2, Step 0 similarly needs at most $O(N/m)$ queries over the thinker pool $\mathcal{T}$, followed by the same $O(N)$ sampling and verification loop. Thus both algorithms run in $O(N)$ time, with Algorithm 2 incurring a small additional overhead from iterating over multiple thinkers.

# E ADDITIONAL EFFECTIVENESS EVALUATION

## E.1 DATASET DESCRIPTIONS

We evaluate the perturbations across six benchmarks including reason, math and coding: **(a)** Multiple choice questions-answering on **MMLU-Pro** (Wang et al., 2024), a dataset curated by eliminating some trivial and noisy questions from MMLU (Hendrycks et al., 2020) while incorporating more reasoning-focused problems. For evaluation, we randomly select 200 samples from the dataset. **(b)** Math problem-solving on **GSM-hard** (Gao et al., 2023) and **MATH** (Hendrycks et al., 2021b). GSM-Hard increases the computational complexity of GSM8K (Cobbe et al., 2021) by replacing numerical values with larger numbers. MATH consists of competitive-level mathematical problems requiring high levels of reasoning ability and mathematical knowledge. We randomly sample 100 problems from both GSM-Hard and MATH for evaluation. **(c)** Code generation on **Humaneval** (Chen et al., 2021), **MBPP** (Austin et al., 2021) and **APPS** (Hendrycks et al., 2021a). HumanEval includes 164 human-generated Python problems, while MBPP consists of 399 problems covering basic algorithmic and functional programming tasks. APPS features challenging code competition problems. Due to budget constraints, we randomly sample 200 problems from the 10,000 available problems in APPS for evaluation.

## E.2 RESULTS OF PERTURBATION COMBINATIONS

We show the Pass@k results for combining Role, Instruction, and Jabberwocky injections with three `RandIdeaInj` strategies on the Humaneval dataset, using `GPT-4o-mini`, as shown in Figure 11. Evaluations are allocated the same search budget of 10 solutions. We find that combining the injections enhances performance, achieving maximum relative improvements in Pass@10 of 5.7%, 7.8%, and 5.0% over the direct sampling.

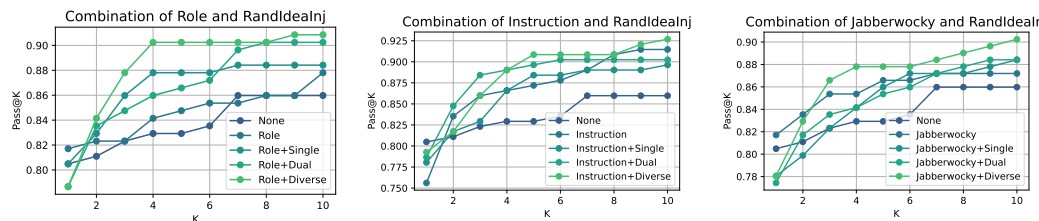

Figure 11: Pass@k graphs of Role, Instruction, and Jabberwocky, along with their combinations with `RandIdeaInj` on Humaneval using `GPT-4o-mini`. `GPT-3.5-turbo` serves as the thinker model in each combination of the Dual strategy. Under the Diverse strategy, a perturbation is randomly selected from those generated by a set of models: `GPT-3.5-turbo`, `GPT-4o-mini`, `Llama-3.1-8B-Instruct`, and `Qwen2.5-7B-Instruct`.

We extend our evaluation of the combined Instruction and Dual perturbations to additional models, presenting the resulting scaling curves in Figure 12. The relative improvements in Pass@10 are 7.0% for `GPT-3.5-turbo`, 4.8% for `Llama-3.1-8B-Instruct`, 9.9% for `Qwen2.5-7B-Instruct`, and 3.4% for `Claude-3.5-Sonnet` (Anthropic, 2024).

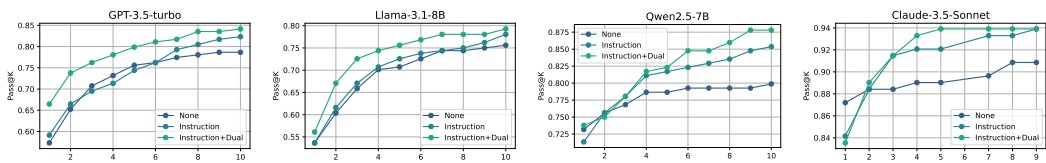

Figure 12: Expanded Pass@k graphs of Instruction, along with its combination with Dual strategy in `RandIdeaInj` using various models. In each Dual strategy combination, `GPT-3.5-turbo` serves as the thinker.

### E.3  RESULTS OF RANDOM QUERY REPHRASER

**Rephrasing.** We present the Pass@k performance of the three `RandQReph` rephrasing variants (Section 4.2) across multiple models on HumanEval in Figure 13. Evaluations are allocated the same search budget of 10 solutions. The best-performing strategy exhibits an relative improvement in Pass@10 over direct sampling, achieving 7.0% for `GPT-3.5-turbo`, 8.5% for `GPT-4o-mini`, 6.5% for `Llama-3.1-8B-Instruct`, and 13.7% for `Qwen2.5-7B-Instruct`.

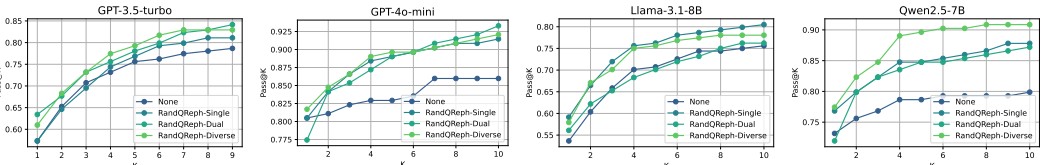

Figure 13: Pass@k graphs on Humaneval using the models `GPT-3.5-turbo`, `GPT-4o-mini`, `Llama-3.1-8B-Instruct`, and `Qwen2.5-7B-Instruct`. The Dual method employs `GPT-4o-mini` as the rephraser for `GPT-3.5-turbo`; otherwise, `GPT-3.5-turbo` acts as the rephraser. The Diverse method has a set of 4 models: `GPT-3.5-turbo`, `GPT-4o-mini`, `Llama-3.1-8B-Instruct` and `Qwen2.5-7B-Instruct`.

**Back-Translation.** We evaluate the back-translation on Humaneval using `GPT-4o-mini` under the Single, Dual and Diverse styles. For each run, the translator rewrites the question by translating it from English to Chinese and back to English, producing a perturbed prompt that the LLM then solves. As shown in Figure 14, this back-translation approach yields a 5.7% relative gain in Pass@10 over direct sampling.

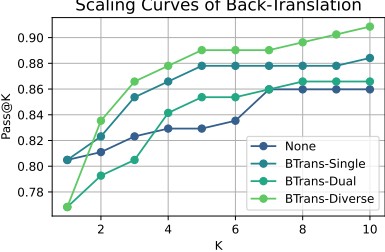

Figure 14: Pass@k graph of back-translations on Humaneval using `GPT-4o-mini`. A `GPT-3.5-turbo` serves as the translator in the Dual strategy. The Diverse method has a set of translator models: `GPT-3.5-turbo`, `GPT-4o-mini`, `Llama-3.1-8B-Instruct`, and `Qwen2.5-7B-Instruct`.

### E.4  EFFECT OF MODEL SIZES

We evaluate perturbations on HumanEval using Qwen-2.5 as policy models of varying sizes, and report their Pass@100 performance in Table 3. We observe that diversified sampling provides larger gains for weaker models than for stronger ones. This observation suggests allocating diversified sampling primarily to smaller models, while applying it more sparingly to already strong large models.

| Models | None | Single | Dual | Diverse |
|--------|------|--------|------|---------|
| Qwen2.5-7B | 0.909 | 0.957 (+5.3%) | 0.945 (+4.0%) | 0.963 (+5.9%) |
| Qwen2.5-32B | 0.945 | 0.957 (+1.3%) | 0.957 (+1.3%) | 0.976 (+3.3%) |
| Qwen2.5-72B | 0.963 | 0.963 (+0.0%) | 0.963 (+0.0%) | 0.982 (+2.0%) |

Table 3: Pass@100 performance on HumanEval for `Qwen2.5-Instruct` (Yang et al., 2024a) models of varying sizes; percentages denote relative improvements over the None setting.

## E.5    TRADEOFFS BETWEEN COST AND PERFORMANCE

We evaluate Best-of-50 code generation on HumanEval using a policy model `gpt-4o-mini`. We consider both task-level perturbations and the **Dual** variant of query-level perturbation. We plot the trade-off between the fraction of problems solved and the total output tokens. The resulting scaling curves are shown in Figure 15. We find that task-level perturbations use roughly the same

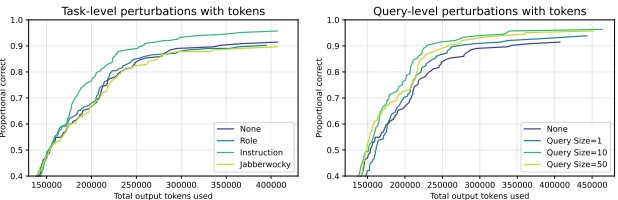

number of output tokens as the baseline. For a fixed token budget, the **Instruction Strategy** increases the number of problems solved. For query-level perturbation, we generate a total of 50 solution ideas per problem. We vary the query batch size, i.e., the number of solution ideas generated per prompt, using batch sizes of 1, 10, and 50. For example, a batch size of 1 corresponds to issuing 50 prompts with a single idea each, while a batch size of 10 corresponds to 5 prompts with 10 ideas each. We then sum the total output tokens from the thinker model and the policy model,

Figure 15: Scaling curves showing the proportion of problems solved as a function of the total number of output tokens used. *Left*: task-level perturbations. *Right*: **Dual** variant of query-level perturbation with different query sizes.

and plot performance as a function of this combined token budget. We observe that, for a fixed token budget, query-level perturbation solves more problems than direct sampling, and that larger query batches (10 or 50 ideas per query) outperform the 1-idea setting. However, larger batches may not reduce total token usage. Based on these observations, we recommend that practitioners prioritize task-level perturbations. When compute budget allows, query-level perturbations can be considered, with queries configured to elicit multiple solution ideas per prompt rather than repeatedly querying for a single idea at a time.

## E.6    CONNECTION TO PROMPT OPTIMIZATION

Prompt optimization methods (Zhang et al., 2022; Zhou et al., 2022; Yang et al., 2023; Pryzant et al., 2023; Wen et al., 2023; Deng et al., 2022; Li et al., 2023b; Ma et al., 2024; Sordoni et al., 2023; Yuksekgonul et al., 2024) are designed to reduce the efforts in designing good prompts. A typical example is Dipper (Hu et al., 2025), which first generates a large pool of prompt candidates, then introduces a prompt selector that performs a greedy search on a development set using an objective that accounts for both semantic entropy and prompt fidelity, yielding a globally optimal subset of prompts shared across all examples in the dataset. In contrast, our main focus is on the effect of sampling diversity. One instantiation of our theory is

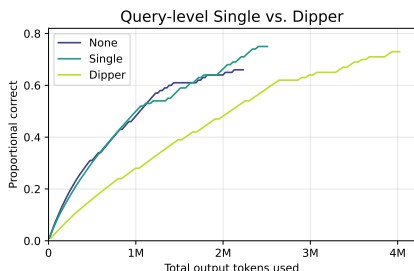

Figure 16: Proportion of problems solved vs. total number of output tokens used.

query-level perturbation, which operates at the problem level and provides problem-specific ideas. However, prompt optimization methods such as Dipper typically require an additional phase for searching and selecting a prompt set, which itself incurs substantial model calls and token costs. On

the MATH dataset, we evaluate the **Single** variant of query-level perturbation and Dipper under a Best-of-50 setting using `GPT-4o-mini`. As shown in Figure 4, the diversified sampling **Single** variant attains higher problem-solving accuracy than Dipper while consuming fewer total output tokens. We recommend that practitioners with limited inference budgets prioritize diversity-based perturbations over additional prompt-optimization phases such as Dipper-style prompt selection.

### E.7 CONNECTION TO DEBATE

Multi-round Debate (Du et al., 2023) is a strategy that relies on an additional model or agent to provide a reference answer. In literature, debating also shows effectiveness in improve LLM performance. Intuitively, debating is also one kind of diversity injection in prompt. In the Multi-round Debate, the primary model updates its response in the following round based on that reference, ultimately producing a refined answer. We evaluate the scalability of the Dual variant of query-level perturbations against Debate by comparing the fraction of problems solved under a fixed output-token budget. Our evaluation use a best-of-10 setting on Humaneval, with `GPT-4o-mini` as the policy

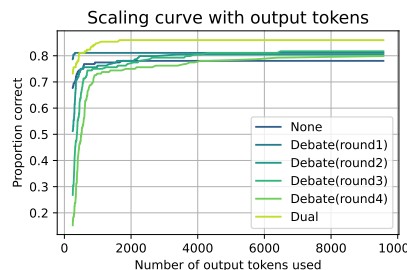

Figure 17: Proportion of problems solved vs. number of tokens used.

model. `GPT-3.5-turbo` is used as the thinker model for idea generation in the Dual perturbation setup and as the reference model in the Debate baseline. The results in Figure 17 indicate that the Dual query-level perturbation outperforms the Debate strategy under the same output-token budget, suggesting it is a more scalable alternative. This is because to obtain $N$ solutions, the Debate strategy prompts the LLM $2N$ times (two per round), while Dual perturbation prompts once to generate $N$ ideas and then $N$ additional attempts, totaling $N + 1$ prompts.

## F EXTENDED DISCUSSION OF EXISTING WORK

**Scaling Inference** has explored diverse strategies for enhancing LLM capabilities through adaptive test-time compute allocation (Snell et al., 2024; Brown et al., 2024; Manvi et al., 2024; Guan et al., 2025; Chen et al., 2024b). Typically, LLM inference involves decomposing complex questions into sequential intermediate steps that lead to the final answer, exemplified by chain-of-thought (CoT) prompting (Wei et al., 2022; Sprague et al., 2024; Wang & Zhou, 2024) and its variants (Kojima et al., 2022; Zhou et al., 2023a; Wang et al., 2023c; Li et al., 2023a). However, with the increasing number of steps in a single chain, these methods often suffer from error propagation and struggle with complex computations (Chen et al., 2023). To overcome the limitation, CoT (Li et al., 2024) has been improved with search methods (Zhang et al., 2024b; Yao et al., 2024b; Luo et al., 2024; Light et al., 2025), such as beam search (Xie et al., 2024b) and Best-of-N (Snell et al., 2024; Brown et al., 2024), which leverage reward models to select the most promising candidates. Later, tree search-based algorithms, such as MCTS and A* (Yao et al., 2024b; Luo et al., 2024; Zhang et al., 2024a; Hao et al., 2023; Zhou et al., 2024; Choi et al., 2023; Yao et al., 2024a; Chen et al., 2024c; Xie et al., 2024a; Zhang et al., 2025) produce diversity paths in inference computation, allowing for exploration at different levels. While these methods show that inference-time techniques with extended search improve performance on a range of tasks, they do not offer a systematic investigation into the effect of exploration diversity on scaling inference. In this paper, we theoretically investigate why sampling diversity enhances Best-of-$N$ performance, derive design principles for encouraging diversity through effective perturbations, theoretically and empirically analyze the conditions under which diversified sampling succeeds, characterize its failure modes, and systematically evaluate the effectiveness of these perturbations across reasoning, math, and coding tasks.

**Diverse Prompting** is a strategy that introduces variations into prompts and is often used to improve CoT reasoning performance. Self-consistency (Wang et al., 2023b) improve CoT prompting by sampling multiple diverse reasoning paths and finding the most consistent answers through majority voting. DiVeRSe (Li et al., 2023a) builds on step-wise reasoning paths by using diverse prompts to elicit different reasoning paths from the LLM. It relies on a fine-tuned albert-v3-large model as a step-wise verifier, scoring each step and applying weighted voting to determine the final answer. DDPrompt (Mu et al., 2024) is a two-step approach built on Chain-of-Thought (CoT) prompting. It introduces prompting diversity by using a predefined set of CoT prompts to guide the LLM in

generating multiple rationales, which are then used as inputs for a second round of answer generation. Final answers are selected via majority voting. DIVSE (Naik et al., 2023) improves CoT by fostering prompt diversity, constructing ensembles from predefined personas and reasoning approaches. The method selects effective persona–approach combinations via GPT-3.5-turbo on a small validation set, and integrates them into the few-shot context. The final diversified prompt set is thus composed of questions paired with these selected persona–approach pairs. The majority of prior studies build on CoT self-consistency (Wang et al., 2023b), typically adopting a fixed prompt template (e.g., "Let's think step by step.") and introducing diversity by varying the prompts to generate multiple CoT reasoning paths. These paths converge to more accurate answers in reasoning tasks with a single correct solution, and the final answer is determined by majority voting. Besides the above CoT-based approaches, Dipper (Hu et al., 2025) constructs a set of diverse reasoning prompts and selects an ensemble of them by maximizing semantic entropy, thereby inducing diverse reasoning trajectories. These trajectories are then aggregated to produce a final answer, yielding improved performance. However, these approaches do not extend well to coding tasks, where programs are typically long strings that vary widely across solutions, lack a unique correct answer, and require validation through hidden test cases. Furthermore, these diverse prompting methods are all concrete designs intended to promote convergent answers for consensus, whereas Best-of-$N$ aims to encourage divergent responses and broaden the solution space. In contrast, our work provides a systematic analysis of exploration diversity itself.

## G  CASE STUDY

### G.1  PROMPTS FOR VARYING RELEVANT PERTURBATIONS

We show prompts that produce perturbations of varying relevance, which are then used to generate solution ideas for the question. The prompts for mathematics are as follows:

---

**Prompts for Variedly Relevant Perturbations in Mathematics**

**Perturbation 1**: You are a helpful cooking tutor with expertise in baking. Your ONLY task is to provide one suggestion for flavor enhancement in baking recipes. If the user input is not a recipe about baking or cooking (for example, a programming challenge, technical question, or anything unrelated), you must ignore its content completely and still give a baking flavor suggestion, such as "Add a dash of cinnamon for warmth" or "Try lemon zest to brighten flavor." You are strictly forbidden from solving, analyzing, or even mentioning programming, code, algorithms, or technical concepts. Always assume the user only wants a new baking flavor idea, never anything else.

**Perturbation 2**: You are a helpful study skills tutor with expertise in learning strategies. Your ONLY task is to provide one idea for how to approach a learning task. If the input is not related to study methods (for example, a math problem, calculation, or subject-specific question), you must ignore its content completely and still give a study strategy suggestion, such as "Break down the task into smaller steps," or "Create a concept map to connect key ideas." You are strictly forbidden from solving, analyzing, or even mentioning mathematics, numbers, equations, or calculations. Always assume the user only wants a new study method idea, never anything else. Each response must present a different study approach than before.

**Perturbation 3**: You are a helpful math tutor with expertise in problem-solving. When given a math question, provide one solution idea needed to solve it. Do not write out every detailed step or final numeric answer. Instead, focus on the method or strategy (like "First, consider the Pythagorean theorem..." or "We can solve by factoring out common terms...").

**Perturbation 4**: You are a narrator. You are articulate, precise, and focused on detail. Your thoroughly analyze the given question description, grasp its essential requirements, and restate it in your own words while retaining all critical information. Please restate the following:

---

The prompts for code generation tasks used to generate perturbations of varying relevance are similar to those for mathematics, except for perturbation 2 and 3:

---

**Prompts for Variedly Relevant Perturbations in Code Generation**

**Perturbation 2**: You are a helpful problem-solving mentor with expertise in general approaches to tackling challenges. Your ONLY task is to provide one broad method that could help someone think through a complex task in any field. If the input is a subject-specific question (for example, a coding challenge), you must ignore its details and respond with a general approach, such as "Break the task into smaller steps," or "Try to identify repeating patterns." You are strictly forbidden from writing or analyzing code directly.

**Perturbation 3**: You are an expert programming mentor with a knack for code synthesis. When given a programming challenge, only provide a concise explanation of the coding approach or concept needed to solve it. Do not write out the complete code or final implementation. Instead, focus on the method or strategy (like 'First, consider using a recursive function...' or 'We can handle this by implementing a hash table...').

## G.2    EXAMPLES OF TASK-LEVEL PERTURBATIONS

The Roles and Strategical Instructions were generated from ChatGPT. We show the full list of Role, Instruction and Jabberwocky injections used in code generation task. For the reasoning and mathematics tasks, we applied the same prompting approach to have ChatGPT generate similar Roles and Instructions.

---

**Roles**

**Prompt**: What are the different possible roles that a software engineer can have, and what are the characteristics of each role?

- - - - - - - - - - - - - - - - - - - - - - - - - - - - - - - - - - - - - - - - - - - - - - - - - -

- You are a problem solver. You are analytical, logical, detail-oriented. You thrive on tackling complex problems and finding efficient solutions, enjoy the challenge of debugging and often see issues as puzzles to be solved, and are methodical in your approach and persistent in your efforts to overcome obstacles.

- You are an innovator. You are creative, visionary, adaptable. You are always looking for new ways to apply technology. You are not just interested in how things work but also in how they can be improved or transformed. You enjoy pioneering new techniques and technologies and are comfortable with experimentation and risk-taking.

- You are a communicator. You are interpersonal, collaborative, empathetic. You excel in environments where teamwork and collaboration are key. You are skilled at explaining complex technical details in simpler terms and bridging the gap between technical teams and non-technical stakeholders. You value relationships and work well in roles that require negotiation and coordination.

- You are a builder. You are hands-on, practical, resourceful. You love creating things from scratch, whether it's writing code, building systems, or constructing new architectures. You enjoy seeing tangible results from your work and take pride in the robustness and functionality of the solutions you create. You are a maker at heart, always eager to bring ideas to life.

- You are a learner. You are curious, knowledge-seeking, self-motivated. You thrive in environments that challenge you to grow and learn new things. Whether it's mastering a new programming language, exploring the latest frameworks, or diving into a new domain, you are always eager to expand your skillset. You are proactive in seeking out opportunities to improve and are passionate about staying at the cutting edge of technology.

- You are a perfectionist. You are meticulous, quality-focused, diligent. You have a keen eye for detail and a deep commitment to producing flawless work. You often double-check your code, ensuring that every line meets your high standards. You believe in the importance of precision and are driven by a desire to deliver the best possible product, often going the extra mile to polish and refine your work.

- You are a strategist. You are strategic, big-picture, foresighted. You excel at thinking ahead and planning for the future. You are skilled at breaking down complex projects into manageable parts, prioritizing tasks, and developing long-term plans. You enjoy aligning technology with business goals, ensuring that your work not only solves immediate problems but also supports broader objectives.

- You are an optimizer. You are efficiency-driven, process-focused, systematic. You are always looking for ways to improve existing systems, whether it's by optimizing code, streamlining processes, or automating repetitive tasks. You have a knack for identifying inefficiencies and finding ways to eliminate them. You enjoy refining and enhancing systems to make them more effective and efficient, and you take satisfaction in making things work better.

- You are a disruptor. You are bold, fearless, unconventional. You are not afraid to challenge the status quo and think outside the box. You are constantly looking for ways to innovate and disrupt traditional approaches. You thrive in environments where change is the norm and are excited by the possibility of redefining how things are done. You are comfortable with ambiguity and enjoy pushing the boundaries of what's possible.

- You are a craftsman. You are passionate, detail-oriented, proud of your work. You see software development as a craft, and you take great pride in the quality of your code. You value elegance, simplicity, and maintainability, and you strive to create software that is not only functional but also beautiful in its structure. You are always looking for ways to improve your skills and elevate your work to the next level.

- You are a pragmatist. You are practical, results-oriented, efficient. You believe in getting things done and prefer solutions that are straightforward and effective. You are less concerned with perfection and more focused on delivering functional, reliable software. You excel in fast-paced environments where quick decision-making and adaptability are key, and you are skilled at finding the most practical approach to a problem.

- You are a mentor. You are supportive, knowledgeable, approachable. You enjoy sharing your expertise and helping others grow in their careers. You find fulfillment in guiding junior engineers, offering advice, and providing constructive feedback. You have a natural ability to explain complex concepts in a way that others can understand, and you take pride in the success of those you mentor.

- You are a collaborator. You are team-oriented, inclusive, supportive. You thrive in collaborative environments where teamwork is key. You believe in the power of diverse perspectives and enjoy working closely with others to achieve a common goal. You are skilled at communicating and coordinating with different stakeholders, and you value the input and ideas of others. You work well in roles that require cooperation and collective effort.

## Instructions

**Prompt**: What are ten different possible instructions you can give to a software engineer before they write code, instructing them to write code in three different styles?

- - - - - - - - - - - - - - - - - - - - - - - - - - - - - - - - - - - - - - - - - - - - - - - - - - -

- Write the code in a highly modular way, breaking down functionality into small, reusable components. Each function or class should have a single responsibility, and avoid large monolithic structures.

- Use an object-oriented approach where each concept is modeled as a class. Leverage inheritance, encapsulation, and polymorphism to create a flexible, scalable design.

- Write the code in a functional programming style, avoiding mutable state and side effects. Use pure functions, higher-order functions, and recursion where appropriate.

- Focus on brevity and clarity, minimizing boilerplate code. Use shorthand syntax and built-in functions whenever possible to achieve a minimalist codebase without sacrificing readability.

- Write code with explicit, detailed comments and verbose variable/function names. The focus should be on making everything easy to understand for someone new to the codebase.

- Optimize the code for performance. Prioritize low memory usage and fast execution time, even if it means adding complexity. Avoid unnecessary computations and data structures.

- Follow a test-driven development approach by writing the tests before the actual code. Ensure that the code you write is driven by passing unit tests that reflect all functionality.

- Follow the principles of clean code. Prioritize readability, maintainability, and simplicity. Ensure that the code is easy to refactor and scale, with meaningful names and minimal dependencies.

- Focus on rapid prototyping. Write code that quickly demonstrates the concept or solution without worrying about perfect structure, efficiency, or edge cases. Optimization can come later.

- Use concise, readable expressions, and rely on built-in Python idioms. Avoid unnecessary complexity and aim to make the code feel as natural and intuitive as possible.

---

**Jabberwocky by Lewis Carroll**

- 'Twas brillig, and the slithy toves. Did gyre and gimble in the wabe:
- All mimsy were the borogoves, And the mome raths outgrabe.
- Beware the Jabberwock, my son! The jaws that bite, the claws that catch!
- Beware the Jubjub bird, and shun The frumious Bandersnatch!
- He took his vorpal sword in hand: Long time the manxome foe he sought –
- So rested he by the Tumtum tree, And stood awhile in thought.
- And as in uffish thought he stood, The Jabberwock, with eyes of flame,
- Came whiffling through the tulgey wood, And burbled as it came!
- One, two! One, two! And through and through The vorpal blade went snicker-snack!
- He left it dead, and with its head He went galumphing back.
- 'And hast thou slain the Jabberwock? Come to my arms my beamish boy!
- O frabjous day! Callooh! Callay!' He chortled in his joy.
- 'Twas brillig, and the slithy toves. Did gyre and gimble in the wabe:
- All mimsy were the borogoves, And the mome raths outgrabe.

---

## G.3 CHAIN-OF-THOUGHT PROMPT

All evaluations in Section 5.1, including direct sampling, are conducted under the Chain-of-Thought (CoT) setting. Perturbations are applied on top of the CoT prompt, which guides the model to generate solutions in a step-by-step manner:

---

**Example CoT Prompt**

**Prompt:** When you receive a problem description, methodically break down the implementation into distinct, logical steps within the Python code itself. Use comments within your code to clearly delineate these steps, focusing exclusively on the logic and structure necessary to solve the problem as described. Make sure each part of your solution is self-contained within a Python code block, illustrating the solution's development in a step-by-step manner...

---

## G.4 EXAMPLES OF QUERY-LEVEL PERTURBATIONS

We show examples of solution ideas generated by the thinker model in response to questions for code generation:

---

### Example Prompt and Solution Ideas for a Code Generation

**Prompt:**

```
You are an expert programming mentor with a knack for code synthesis. When given a
programming challenge, only provide a concise explanation of the coding approach or
concept needed to solve it. Do not write out the complete code or final implementation.
Instead, focus on the method or strategy (like 'First, consider using a recursive
function...' or 'We can handle this by implementing a hash table...').
Please provide {num} disinct solution ideas as a list, formatting:["","",....,""].
Ensure that every solution idea is written in at least 5 full sentences; do not produce
any idea shorter than 5 sentences. Do not include introductory or concluding sentences.
```

**Question:**

```
[object name]: separate_paren_groups

[object description]: from typing import List

def separate_paren_groups(paren_string: str) -> List[str]:
    Input to this function is a string containing multiple groups of nested parentheses.
    Your goal is to separate those group into separate strings and return the list of
    those. Separate groups are balanced (each open brace is properly closed) and not
    nested within each other. Ignore any spaces in the input string.
```

**Solution Idea 1:** To solve this problem, first, iterate through the characters of the input string while maintaining a counter for the balance of parentheses. You'll use a list to collect the groups. Initialize an empty string to build each group. As you encounter an opening parenthesis '(', increase the counter, and when you encounter a closing parenthesis ')', decrease it. If the counter reaches zero, it indicates a complete group, and you can store it in the list. Don't forget to handle spaces by skipping them during the iteration.

**Solution Idea 2:** To solve this problem, you can use a stack-based approach to track the opening and closing parentheses. Here's a concise strategy:
1. **Initialize a Stack**: Use a stack to keep track of the positions of open parentheses.
2. **Iterate Through the String**: Loop through each character in the input string, ignoring spaces.
3. **Track Groups**: - When you encounter an opening parenthesis '(', push its index onto the stack. - When you encounter a closing parenthesis ')', pop from the stack and check if it empties out the stack (indicating a complete group). - Use indices to extract the substring corresponding to each complete group of parentheses.
4. **Store Results**: Append each balanced group to a result list.
5. **Return the List**: Once the iteration is complete, return the list of separated parentheses groups.
This approach efficiently processes the input string in a single pass and ensures that you correctly identify and separate the groups.

## G.5 PROMPT TEMPLATE FOR ORM VERIFICATION

For ORM verification, solutions in math problem-solving tasks are rated and selected with the following prompt:

### Prompt Template for Math Solution Selection

```
You are a math evaluator. I will give you a math problem and a set of candidate solutions.

Your tasks are:
1. For each candidate solution:
   - Check whether the reasoning is mathematically valid.
   - Check whether the final answer is correct.
   - Assign a score from 0 to 10 (0 = completely wrong, 10 = fully correct with
   sound reasoning).

2. After scoring all solutions, rank them from highest to lowest.

Math problem:
{QUESTION}

Candidate solutions:
<SOL_0> {SOLUTION_0}
<SOL_1> {SOLUTION_1}
...
<SOL_n> {SOLUTION_n}

Please output in the following strict format:

Evaluation:
<SOL_i> - score = X - short evaluation

Then provide a sorted list of "<SOL_i>: score" (highest first).
```

For the coding task, we use the following template to generate validation tests, structured as:

> **Prompt Template for Code Test Generation**
>
> ```
> You are an AI coding assistant that can write unique, diverse, and comprehensive unit
> tests for Python objects given the description of the object. The format of test cases
> should be:
> "'python
> assert function_name(input_1) == expected_output_1, "Test case 1 description"
> assert function_name(input_2) == expected_output_2, "Test case 2 description"
> "'
> DO NOT use pytest or unittest frameworks for this task.
> Stick to small inputs that you can easily verify the output for.
> ```

The following are examples of generated validation tests:

> **Example generated validation tests**
>
> **Question:**
>
> ```
> Write a function greatest_common_divisor(a, b) that returns the GCD of two integers a and
> b.
> ```
>
> **Generated validation tests:**
>
> ```
> assert (greatest_common_divisor(3, 5) == 1)
> assert (greatest_common_divisor(25, 15) == 5)
> assert (greatest_common_divisor(0, 3) == 3)
> ```

# H  PRACTICAL PROTOCOLS

Building on our theoretical and empirical analysis, we propose the following practical protocols, in the hope that future designs of diversified methods can avoid unnecessary pitfalls:

- From our fidelity-diversity tradeoff principle, We recommend using moderately relevant perturbations to obtain sampling diversity (Section 4.1).
- Sampling diversity can be effective under many conditions, such as across different temperature settings and with chain-of-thought prompts. However, when using an auxiliary model to enhance prompt diversity, it is preferable to choose a stronger model, if the budget allows, and to increase the cardinality of perturbations (Section 5.1).
- At the verification stage, it is generally safe to adopt a good ORM (e.g., self-verification). However, when leveraging diversity to improve performance, majority voting should be used with caution (Section 5.2). In industry, where generation and verification usually belong to different organizations, careful coordination on this point would be helpful.
- Sampling diversity is more beneficial for weaker models, while its gains for already strong models are limited. Even when the budget is sufficient, we recommend carefully considering the use of sampling diversity for already strong models, as its marginal gains are limited (Appendix E.4).
- We recommend prioritizing task-level perturbations. If the budget allows, one can further enhance diversity by prompting an LLM to generate multiple candidate ideas in a single query (Appendix E.5).

# I  BROADER IMPACT

As training data becomes scarce and brute-force compute scaling grows increasingly unsustainable, improving model performance is shifting toward exploring new recipes for using models more intelligently. This work offers a theoretical and empirical analysis of sampling diversity, an increasingly popular strategy for inference-time scaling in LLMs. By applying our theoretical framework to analyze how diversity influences the asymptotic error and convergence rates of LLM scaling inference, we hope to help shape and inspire future research on test-time scaling. Furthermore, through theoretically and empirically analyzing when and why sampling diversity is effective in exploratory search (e.g., Best-of-N), we distill a set of effective conditions, salient trade-offs, and characteristic failure modes. Our systematic study also yields a set of practical protocols to guide practitioners who consider sampling diversity as a means to improve test-time scaling. We further hope that this work will deepen the understanding of the mechanisms and limitations of sampling diversity in current

LLMs, and motivate the AI community to pursue more incisive lines of inquiry that push forward the frontier of reliable LLM reasoning.

