# OpenReview forum: "On the Effect of Sampling Diversity in Scaling LLM Inference"
_ICLR.cc/2026/Conference — Submitted to ICLR 2026_

### Official Review · Reviewer_XCHZ · 2025-10-29

**Soundness:** 3
**Presentation:** 3
**Contribution:** 1
**Rating:** 2
**Confidence:** 4

**Summary:**

This work studies the effect of prompt diversity in scaling inference. The authors also explain how diversified sampling helps lower Best-of-N error rates. They conduct a systematic study of different prompt perturbation approaches and notice that the proportion of k correct solutions increases when the produced candidates are more dissimilar. The paper  (1) theoretically demonstrates that prompt perturbations improve best-of-N performance by analyzing “blind spots” and “rates of convergence”, (2) analyzes the range of perturbation that yields positive gains, (3) analyzes conditions under which the gains of diversified sampling are not outdone by other operations and (4) empirical evaluations on reasoning, math and code generation to see which task-level and query-level perturbations have more consistent gains.

**Strengths:**

- The paper conducts a timely analysis and contains useful empirical work.

- The taxnonomy of perturbations is novel. The formalisms are very clear

- Theorem 3.5 is elegant in deadling with two main desiderata of inference-time scaling

- A good set of similarity metrics is used to properly quantify diversity or lack thereof

**Weaknesses:**

### Major
1. The paper reads like a collection of observations without a clear conclusion or even actionable items or tools for the community to benefit from. I am open to being concinved otherwise, but I cannot recommend acceptance given the way the paper reads now.

2. My main concern is novelty, as the conclusions seem self-evident. Findings like a non-linear 'sweet spot' for perturbation are trivial, while noting that diversity-oriented methods fail under majority-voting reward settings is tautologous.

### Minor
3. The meaning of "policy solver mode diversity" is unclear. Please write it more clearly.

4. I only understood what Jabberwocky was after reading the Appendix. Please make it clearer in the main body. See Q1.

5. While the paper is generally on the easier side of parsing, there are very dense paragraphs that use specific **ad-hoc**(paper-specific) concepts that are not common. For example, line **246 to 254** is difficult to parse; I suggest using colors or stylized text. Figure 3 and 4 are also very unclear. The colors are difficult to separate and the line styles hinder the delivery of the empirical insights.

6. Relevant citations regarding empirical analysis of temperature scheduling effects or ways of increasing diversity are missing [1] [2]

[1] Zhu, Alan, et al. "BARE: Leveraging Base Language Models for Few-Shot Synthetic Data Generation." arXiv preprint arXiv:2502.01697 (2025).

[2] Ahmed, Eltayeb, et al. "Intent Factored Generation: Unleashing the Diversity in Your Language Model." arXiv preprint arXiv:2506.09659 (2025).

**Questions:**

### Major
Q1. Why did you choose the Jabberwocky perturbation? What is special about it? If nothing then what makes it so representative of that family of perturbation.

Q2. The empirical evaluation is not above 1.2 temp? Why this arbitrary limitation?

Q3. What is the takeaway from Table 2?

Q4. Have you considered using Semantic Entropy as an alternative metric for diversity? [1]

### Minor
Q4. Is the illustration asset used in the textboxes copyright-free? Please clarify the licenses if you have not created the image yourself. This does not bear any weight on the score. I just want to make sure proper attribution is always given.

[1] Kuhn, Lorenz, Yarin Gal, and Sebastian Farquhar. "Semantic uncertainty: Linguistic invariances for uncertainty estimation in natural language generation." arXiv preprint arXiv:2302.09664 (2023).

---

> ### Author Response · Authors · 2025-11-24
> **Official response to reviewer XCHZ (1/3)**
>
> Dear reviewer XCHZ,
>
> Thanks for your concerns and suggestions. We address your concerns and questions below:
>
> **W1. The paper reads like a collection of observations without a clear conclusion or even actionable items or tools for the community to benefit from. I am open to being concinved otherwise, but I cannot recommend acceptance given the way the paper reads now.**
>
> **Response:**  We thank you for your concern. However, this paper is **not merely an empirical observation or discovery**. Instead, we present a **systematic analysis** of the increasingly popular diversity-based exploration techniques. We believe that as this technique develops, **systematic analysis** becomes necessary. Practitioners may know that the technique works, but often lack a clear understanding of **why** it works, **under what conditions** it is effective, and **how** it should be used.
>
> We derive a number of conclusions, most from a theoretical perspective and further supported by empirical evidence. For better clarity, we have revised our conclusion in the manuscript to emphasize that it presents a systematic analysis. To clarify our contributions, we summarize them as follows:
>
> - We provide a theoretical foundation analyzing why diversity enhanced Best-of-$N$ performance, suggesting that increasing sampling diversity leads to faster convergence as $N\rightarrow \infty$ and a lower asymptotic error. We hope that our theoretical framework will help *shape* and *inspire* future analytical directions for studying scaling inference.
> - Building on our main theorem, we derive a diversity–fidelity trade-off principle that guides the design of sampling strategies. This principle is supported by our empirical observation that moderately meaningful perturbations are particularly effective, while overly similar or irrelevant ones yield no performance gains and can even lead to degradation.
> - We theoretically and empirically analyze **when** diversified exploration remains effective, showing that sampling diversity remain effective across a wide range of conditions, but depend on many factors.
> - Our theoretical proposition shows that majority voting constitutes a failure mode in which diversity does not lead to performance gains, a behavior that is also confirmed empirically, suggesting that diversity cannot be applied indiscriminately.
> - We conduct extensive and systematic evaluations of sampling diversity on reasoning, mathematics, and code generation tasks, quantifying the performance gains it can deliver.
>
> Overall, our systematic analysis provides a **theoretical and empirical
> foundation for understanding the effect of diversity in LLM inference-time scaling.**
>
> **Tools:** As stated, we will release our codebase and data for full reproducibility, including all experiments. We will clean the codebase and provide easy-to-use instructions.
>
> **Actionable items:** Our actionable items can be summarized as follows:
>
> - From our fidelity-diversity tradeoff principle, We recommend using moderately relevant perturbations to obtain sampling diversity (Section 4.1).
> - Sampling diversity can be effective under many conditions, such as across different temperature settings and with chain-of-thought prompts. However, when using an auxiliary model to enhance prompt diversity, it is preferable to choose a stronger model, if the budget allows, and to increase the cardinality of perturbations (Section 5.1).
> - At the verification stage, it is generally safe to adopt a good ORM (e.g., self-verification). However, when leveraging diversity to improve performance, majority voting should be used with caution (Section 5.2). In industry, where generation and verification usually belong to different organizations, careful coordination on this point would be helpful.
> - Sampling diversity is more beneficial for weaker models, while its gains for already strong models are limited. Even when the budget is sufficient, we recommend carefully considering the use of sampling diversity for already strong models, as its marginal gains are limited (Appendix E.4).
> - We recommend prioritizing task-level perturbations. If the budget allows, one can further enhance diversity by prompting an LLM to generate multiple candidate ideas in a single query (Appendix E.5)
>
> In the revised manuscript, we have added Appendix **H**, “Practical Protocols.”

---

> ### Author Response · Authors · 2025-11-24
> **Official response to reviewer XCHZ (2/3)**
>
> **W2. My main concern is novelty, as the conclusions seem self-evident. Findings like a non-linear 'sweet spot' for perturbation are trivial, while noting that diversity-oriented methods fail under majority-voting reward settings is tautologous.**
>
> **Response:** As we stated in our previous response, the goal of this paper is not to present a single new discovery, but rather to provide a **systematic analysis** of the increasingly popular notion of exploration diversity. Our aim is to offer a theoretical and empirical framework for understanding the effect of diversity in LLM inference-time scaling.
>
> In this work, we provide a theoretical framework that explains **why** diversity improves scaling inference (e.g., Best-of-$N$), and from this framework we derive principles that characterize **when** it is effective, why it is effective, and what its failure modes are. Our empirical observations and theoretical analysis mutually reinforce each other: the observations support and confirm the theoretical predictions. We hope that our analytical perspective will provide useful guidance for future work on understanding test-time scaling. We believe that providing such theoretical and empirical analysis belongs to what the academic community aim to do.
>
> In addition, the failure mode of majority voting is one of the aspects considered in our analysis. Our analysis suggests that, if diversity is treated as a key lever for improvement, then majority voting is not an appropriate accompanying strategy. This perspective has been largely overlooked in prior work. We believe it is especially important in industrial contexts, where generation and verification are often handled by different organizations, and better coordination can be beneficial.
>
> While we agree in spirit that our corollaries about the diversity-fidelity trade-off and the majority voting failure mode are not surprising discovery, our theoretical contribution goes beyond these downstream conclusions.
>
> The main conclusion of our theory, as presented in Theorem 3.5, claims a faster convergence rate by a factor of $C_N=\Omega \big( \hat{\mu}_1^2 N/(1+\epsilon) \big)$, providing an exact order of dependency between the convergence increase factor and **diversity quantifier** $\hat{\mu}_1$ and **fidelity quantifier** $\epsilon$. This conclusion, as well as the entire theoretical framework we established to achieve it, help us form provable predictions on the effects of diversified sampling (diversity-fidelity trade-off and majority voting failure), which are further corroborated by our empirical evidence.
>
> In the revised manuscript, we provide a detailed exposition of the diversity–fidelity trade-off principle and the derived failure mode under majority voting in Sections 4.1 and 5.2.
>
> **W3. The meaning of "policy solver mode diversity" is unclear. Please write it more clearly.**
>
> **Response:** Thank you for your careful reading. We have revised our manuscript to simply use “sampling diversity” when talking in general terms, and only mention sampling **mode** $\zeta$ in theoretically rigid discussions within Section 3, where we use it to denote all randomness present in a sampling strategy beyond the base LLM decoding randomness.
>
> **(W4, Q1) I only understood what Jabberwocky was after reading the Appendix. Please make it clearer in the main body. See Q1: Why did you choose the Jabberwocky perturbation? What is special about it? If nothing then what makes it so representative of that family of perturbation.**
>
> **Response:**  Thanks for the thoughtful question. *Jabberwocky* is a classic nonsense poem, often regarded as a prototypical example of meaningless text (nonsense verse) [R1]. We expect similar results if we replace it with other, similarly irrelevant texts. Thus, we use *Jabberwocky* as a representative instance of this family of perturbations, employing irrelevant, random text as a control experiment to test whether arbitrary perturbations can also increase diversity. In our empirical observation, we find that compared with direct sampling, it does not significantly improve solution diversity nor noticeably increase the pass rate, suggesting that irrelevant perturbations are ineffective.
>
> In the revised manuscript, we have added a footnote at the beginning of Section 5 to further describe it.
>
> [R1]. Carroll, L. (1978). *Through the Looking-glass and what Alice Found There*. Doubleday Classics.

---

> ### Author Response · Authors · 2025-11-24
> **Official response to reviewer XCHZ (3/3)**
>
> **W5. While the paper is generally on the easier side of parsing, there are very dense paragraphs that use specific ad-hoc (paper-specific) concepts that are not common. For example, line 246 to 254 is difficult to parse; I suggest using colors or stylized text. Figure 3 and 4 are also very unclear. The colors are difficult to separate and the line styles hinder the delivery of the empirical insights.**
>
> **Response:**  Thanks for your careful reading and constructive suggestions. In the revised manuscript, we have revised Section 5.1 and applied a different font to terms that are used only as specific names within the paper. We have redrawn Figures 3 and 4 and present them separately.
>
> **W6. Relevant citations regarding empirical analysis of temperature scheduling effects or ways of increasing diversity are missing [1] [2]**
>
> **Response:**  Thanks for the constructive suggestion. We added related work [R1, R2] on temperature scheduling to Section 5.1 in the revised manuscript.
>
> [R1]. Zhu, Alan, et al. "BARE: Leveraging Base Language Models for Few-Shot Synthetic Data Generation." arXiv preprint arXiv:2502.01697 (2025).
>
> [R2].Ahmed, Eltayeb, et al. "Intent Factored Generation: Unleashing the Diversity in Your Language Model." arXiv preprint arXiv:2506.09659 (2025).
>
> **Q2. The empirical evaluation is not above 1.2 temp? Why this arbitrary limitation?**
>
> **Response:**  Thanks for your careful reading. In this empirical study, our goal is to understand how sampling diversity behaves across different temperatures, focusing on conclusions that are relevant for practical use. In practice, temperatures substantially above 1.0 are seldom used, as they are typically reported to produce very low-fidelity, almost random outputs and are thus rarely adopted in practical deployments.  We considered the range 0.0–1.2 to already cover and slightly extend the standard operating regime. We additionally include results at temperature 1.4 in Section 5.1 of the revised manuscript.
>
> **Q3. What is the takeaway from Table 2?**
>
> **Response:**  Thanks for your careful reading. Table 2 is to show that higher solution diversity often lead to better performance, which empirically corroborates our Theorem 3.5. We provide a more detailed explanation in the revised manuscript.
>
> **Q4. Have you considered using Semantic Entropy as an alternative metric for diversity? [1]**
>
> **Response:**  Thanks for your insightful suggestions. We integrated this method into our codebase to quantify solution similarity for Best-of-$N$ on MATH and HumanEval. However, we got close to zero results and show no meaningful variation across methods and datasets. It reflects a mismatch between semantic entropy and the types of outputs considered here.
>
> First, semantic entropy was explicitly designed for free-form natural-language QA, where different surface realizations can express the same meaning and a bi-directional NLI model can cluster paraphrases into "semantic classes”. However, math and code solutions are symbolic snippets rather than natural-language answers, and their semantics are not reliably captured by NLI-based semantic entropy. Mathematically equivalent expressions and functionally equivalent programs that use different variable names or control flow are not reliably recognized as equivalent by an NLI model trained on natural language. As a result, the semantic-entropy pipeline either collapses almost all generations into a single “meaning” or produces clusters that do not correspond to true semantic similarity.
>
> Second, the semantic entropy algorithm requires $O(M^{2})$ pairwise NLI calls over M samples per problem. In our setting, we routinely use much large N (e.g. N=100). At this scale, the NLI-based clustering becomes the dominant computational cost and is not practically scalable.
>
> For these reasons, we consider that semantic entropy may not a suitable or scalable similarity measure for math and code generation in our setting. We have therefore not included these near-zero, uninformative numbers as results.
>
> **Q5. Is the illustration asset used in the textboxes copyright-free? Please clarify the licenses if you have not created the image yourself. This does not bear any weight on the score. I just want to make sure proper attribution is always given.**
>
> **Response:**  We sincerely thank you for carefully reading our manuscript and pointing out details that we had overlooked. The icon used in our text box was downloaded from *Flaticon.com*. This icon is provided under the “Free for personal and commercial use with attribution” license. Accordingly, we have added an appropriate attribution statement as a footnote on the last page of the revised manuscript.
>
> Thank you for your careful reading and for raising your concerns and suggestions. We are glad to address your further comments and questions

---

> ### Author Response · Authors · 2025-11-25
> **A sincere and kind reminder to the ICLR Reviewer XCHZ**
>
> Dear ICLR Reviewer XCHZ,
>
> We greatly appreciate your time and the insightful comments provided during the review of our paper. We have made extensive efforts to address all your questions, suggestions, and misunderstandings in the response and believe that they adequately address all your concerns. We are grateful for your constructive suggestions.
>
> **With the discussion phase ending soon, we would like to confirm whether there are any other clarifications they would like. If the reviewer's concerns are clarified, we would be grateful if the reviewer could increase the score.**
>
> Thank you again for your time and valuable input; we are deeply appreciative.
>
> Best regards,
>
> Authors

---

> ### Comment · Reviewer_XCHZ · 2025-11-25
>
> Thank you for the detailed response. I think the paper makes valuable insights. I will leave it to the AC to decide if such an analysis paper is a good fit for the venue. I will update my score accordingly

---

> ### Author Response · Authors · 2025-11-26
> **Follow-up to Reviewer XCHZ**
>
> Dear Reviewer XCHZ,
>
> Thank you for carefully reading our rebuttal and for updating your score. We greatly appreciate your recognition that the paper offers valuable insights. Many of your comments and suggestions led to fruitful discussions and helped us improve the overall quality of the paper. In the updated version, we have also added a “Broader Impact” statement in Appendix I.
>
> We are grateful for your recognition and support of our work.
>
> Sincerely,
>
> Authors

---

### Official Review · Reviewer_BKDX · 2025-10-30

**Soundness:** 3
**Presentation:** 2
**Contribution:** 3
**Rating:** 4
**Confidence:** 4

**Summary:**

The authors investigate the impact of prompt diversity on scaling inference in LLMs. They provide a theoretical explanation for why diversified sampling improves Best-of-N metric performance. The paper also analyzes the perturbation fidelity and demonstrates that moderately relevant perturbations yield the best results, offering practical guidance for designing effective perturbations. Building on this analysis, the authors propose a set of validated perturbation strategies at both the task level and query level, and discuss the conditions under which these approaches succeed. The conclusion is validated through experiments conducted across diverse tasks.

**Strengths:**

**Theoretical explanation.** The paper presents a formal theoretical result showing that diversified sampling reduces the failure probability in Best-of-N inference. This offers a conceptual foundation rather than only relying on empirical results.

**Experimental results.** The authors conduct extensive experiments across diverse tasks and settings, providing strong empirical evidence to support their ideas.

**Practical implications.** The work offers practical insights into designing effective perturbations that enhance LLM inference performance, providing insights and directions for future work and practical applications.

**Weaknesses:**

**1. Missing cost consideration in experiments.**
While the experiments demonstrate that diversified sampling improves LLM performance, the paper does not discuss the associated computational or time costs. An analysis of the trade-off between performance gains and inference cost would strengthen the empirical evaluation and clarify the practical value of the approach.

**2. Connections and readability.**
The paper lacks clear connections between sections—specifically between notation, theoretical analysis, and experimental validation. Providing smoother transitions and contextual links would greatly improve readability and help readers understand how each component correspons to other parts.

**3.Strong assumptions.**
Hypothesis 3.1 (Variation under auxiliary diversity) assumes that, for any fixed input $\bf{r}$, the variation induced by auxiliary diversity is uniformly bounded below by a constant $\widehat{\mu}_1$. This uniform bound may be overly strong and requires further justification in realistic settings. How about the inputs that are already easy or perturbations have little effect?

See also in questions below.

**Questions:**

1. Regarding the hybrid diversity term $\xi \sim \Pi$, what exactly are $\xi$ and $\Pi$ in the proposed strategies and in the experimental setups?

2. The experimental results mainly focus on performance across different tasks. How does the proposed diversified sampling perform across different LLMs, for instance, models of varying sizes?

3. Could the authors provide a more detailed explanation of Hypothesis 3.1 and its practical meaning? (See also Weakness 3)

4. Some minor questions:

     (1) What does ``RNG'' refer to in Remark 3.2 and Hypothesis 3.3?

     (2) How are $N^{K}$ and $N^{\text{inf}}$ defined in Theorem B.5 in Appendix B.4?

---

> ### Author Response · Authors · 2025-11-24
> **Official response to reviewer BKDX (1/2)**
>
> Dear reviewer BKDX,
>
> Thank you for your comments and suggestions. We address your concerns below:
>
> **W1. Missing cost consideration in experiments. While the experiments demonstrate that diversified sampling improves LLM performance, the paper does not discuss the associated computational or time costs. An analysis of the trade-off between performance gains and inference cost would strengthen the empirical evaluation and clarify the practical value of the approach.**
>
> **Response:** Thank you for your insightful question. In the revised manuscript, we have added a time complexity analysis in Appendix **D**. We also evaluated and analyzed the trade-off between token usage and problem-solving accuracy for the task-level and query-level perturbations. The results and discussions are in Appendix **E.5**, “Tradeoffs between Cost and Performance,” in the revised manuscript.
>
>
>
> **W2. Connections and readability. The paper lacks clear connections between sections—specifically between notation, theoretical analysis, and experimental validation. Providing smoother transitions and contextual links would greatly improve readability and help readers understand how each component correspons to other parts.**
>
> **Response:**  We thank the reviewer for constructive suggestions. In the revised manuscript, we have added transitions and connections within each section and subsection. In particular, we introduced connecting sentences throughout Section 3 to provide a clearer roadmap for the theoretical proofs.
>
> **3.Strong assumptions. Hypothesis 3.1 (Variation under auxiliary diversity) assumes that, for any fixed input $r$, the variation induced by auxiliary diversity is uniformly bounded below by a constant $\hat{\mu}_1$. This uniform bound may be overly strong and requires further justification in realistic settings. How about the inputs that are already easy or perturbations have little effect?**
>
> **Response:**  We agree with the reviewer in spirit that the uniform moment lower bound in Hypothesis 3.1 may appear strong. However, the only case where the assumption may become unrealistic is when $q(r,\zeta)\approx 0$, or when the input is close to impossible to solve, to the point that the solver has an almost 0% success rate under any circumstances. For this purpose, we exclude inputs that have $q(r,\zeta) = 0$ in our hypothesis.
>
> In the case where the input is very easy, $q(r, \zeta)$ would be a very low negative quantity, and so analytically any small variation in $\zeta$ would induce a large change, leading in fact to a **larger** first order moment. As an extreme but intuitive example, if some prompt perturbation $\zeta$ leads to a near-perfect model sampling that always produces the correct result, the corresponding $q(r, \zeta) = -\infty$, leading to a first order moment of $\infty$.
>
>
>
> **Q1. Regarding the hybrid diversity term $\xi \sim \Pi$, what exactly are $\xi$ and $ \Pi$ in the proposed strategies and in the experimental setups?**
>
> **Response:**  We thank the reviewer for the fair question. In our experiments, $\xi$ corresponds to the specific prompt perturbation injection used, and $\Pi$ is therefore the distribution of prompt injections. For example, for Jabberwocky injection, $\xi$ denotes the specific nonsensical text added to the prompt, while $\Pi$ denotes a uniform distribution over all possible injection texts.
>
> **Q2. The experimental results mainly focus on performance across different tasks. How does the proposed diversified sampling perform across different LLMs, for instance, models of varying sizes?**
>
> **Response:** Thank you for the constructive suggestion. We evaluate perturbations on HumanEval using Qwen2.5-Instruct of varying sizes and report their Pass@100 performance. We find that diversified sampling yields larger gains for weaker models than for stronger ones. This observation suggests allocating diversified sampling primarily to smaller models, while applying it more sparingly to already strong large models. We have added these results to Appendix E.4 in the revised manuscript.
>
> | Models      | None  | Single | Dual  | Diverse |
> | ----------- | ----- | ------ | ----- | ------- |
> | Qwen2.5-7B  | 0.909 | 0.957  | 0.945 | 0.963   |
> | Qwen2.5-32B | 0.945 | 0.957  | 0.957 | 0.976   |
> | Qwen2.5-72B | 0.963 | 0.963  | 0.963 | 0.982   |

---

> ### Author Response · Authors · 2025-11-24
> **Official response to reviewer BKDX (2/2)**
>
> **Q3. Could the authors provide a more detailed explanation of Hypothesis 3.1 and its practical meaning? (See also Weakness 3)**
>
> **Response:** We thank the reivewer for careful reading. For both Hypothesis 3.1 and Hypothesis 3.3, we enriched their respective remarks to better explain their intuitions and practical meanings:
>
> **Remark 3.2** (Intuition for Hypothesis 3.1)
>
> Unless $r$ is truly unsolvable for *all* $(\eta,\xi)$, varying the auxiliary randomness $\xi$ induces variation into the success probability. This is because a large auxiliary diversity (e.g. prompt perturbation) can lead to aggressively different decoding paths compared to base policy, which can then lead to larger variations in success probabilities. Note that averaging over $\eta$ reflects that we compare attempts under the same baseline stochasticity.
>
> **Remark 3.4** (Intuition for Hypothesis 3.3)
>
> Here the auxiliary choice $\xi$ is *fixed* on both sides so that both policies enjoy the same amount of base variability ($\eta\sim\nu_0$). Fidelity requires that switching to a new auxiliary setting $\xi^\star$ does not globally worsen the response quality compared to a reference $\xi^0$. It is worth noting that Hypothesis 3.3 should be considered a guideline for selecting good auxiliary diversity sources rather than an assumption: good decoding modes should preserve the fidelity of generation results. For example, with prompt perturbation, one should first test if a specific perturbation method degrades generation quality before applying it as part of the diversity suite.
>
> **Q4. Some minor questions: (1) What does ``RNG'' refer to in Remark 3.2 and Hypothesis 3.3? (2) How are and defined in Theorem B.5 in Appendix B.4?**
>
> **Response:** Thank you for careful reading of our paper. We answer the questions as follows:
>
> (1) Since the term “RNG” was never defined in our original manuscript, we have opted to change it into “randomness” for simplicity.
>
> (2) We apologize for the typos in notation; the terms $N^K$ and $N^{\text{inf}}$ are supposed to be $P^K$ and $P^{\text{inf}}$, defined in equations (4) and (5) in our main texts. We have corrected these notations and restated their definitions in Appendix B.
>
> Thank you for taking the time to carefully read our paper and for raising detailed concerns and questions. We hope our responses have addressed these points, and we are pleased to respond to any further questions or comments you may have.

---

> ### Author Response · Authors · 2025-11-25
> **A sincere and kind reminder to the ICLR Reviewer BKDX**
>
> Dear ICLR Reviewer BKDX,
>
> We greatly appreciate your time and the insightful comments provided during the review of our paper. We have made extensive efforts to address all your questions, suggestions, and misunderstandings in the response and believe that they adequately address all your concerns. We are grateful for your constructive suggestions.
>
> **With the discussion phase ending soon, we would like to confirm whether there are any other clarifications they would like. If the reviewer's concerns are clarified, we would be grateful if the reviewer could increase the score.**
>
> Thank you again for your time and valuable input; we are deeply appreciative.
>
> Best regards,
>
> Authors

---

> > ### Author Response · Authors · 2025-11-26
> > **Kind reminder to Reviewer BKDX (discussion deadline is approaching)**
> >
> > Dear Reviewer BKDX,
> >
> > Thank you again for your time and efforts in reviewing our paper. We have carefully responded to each of your questions.
> >
> > Given that the author-reviewer discussion deadline is approaching, we would greatly appreciate it if you could kindly review our responses and share your valuable feedback. We would be grateful if you could consider increasing the score given our clarification.
> >
> > Thank you very much!
> >
> > Best regards,
> >
> > Authors

---

> ### Author Response · Authors · 2025-11-27
> **Kind reminder to Reviewer BKDX**
>
> Dear Reviewer BKDX,
>
> We greatly appreciate your time and the insightful comments provided during the review of our paper.
>
> We have made extensive efforts to address all your questions, suggestions, and misunderstandings in our response and believe that they adequately address your concerns. The reviewers’ comments focused primarily on providing a cost analysis, improving the connections and transitions in the writing, and clarifying certain hypotheses and theorem. We have addressed these points by strengthening connections and transitions in the manuscript, providing more detailed explanations (including clarifications of specific claims), and adding additional results.
>
> As you and other reviewers have noted, our work offers valuable insights, providing a systematic analysis that offers a theoretical and practical foundation for understanding the sampling diversity in LLM inference-time scaling. Since your comments do not impact the major contributions of our manuscript, we would be grateful if you could consider increasing the score.
>
> We are extremely grateful for your time.
>
> Best regards,
>
> The Authors

---

### Official Review · Reviewer_kUBb · 2025-11-01

**Soundness:** 2
**Presentation:** 1
**Contribution:** 2
**Rating:** 2
**Confidence:** 3

**Summary:**

This paper shows that sampling diversity, varying prompts instead of reusing the same one, improves Best-of-N performance in LLM inference. The authors theoretically prove that diverse sampling reduces errors and speeds convergence, and empirically find that moderately relevant perturbations (neither too similar nor too random) yield the best results. Through task-level and query-level perturbations, they report consistent gains across reasoning, math, and coding tasks, up to 10.8% in EM@100, demonstrating exploration diversity as an effective, test-time strategy for enhancing LLM performance without model changes.

**Strengths:**

1. Introduces a novel and systematic study of sampling diversity in LLM inference, combining theoretical proof, perturbation design, and empirical validation.

2. Demonstrates that exploration diversity can significantly enhance LLM performance without retraining, offering a practical, general-purpose strategy for improving test-time scaling across domains.

**Weaknesses:**

1. The writing and presentation are weak and difficult to follow. For example, in Section 3 (line 144), the term RNG appears without any prior introduction or explanation. Additionally, the two hypotheses are presented without sufficient depth or illustrative support; although Remark 3.2 attempts clarification, it remains unclear. Including concrete examples or clearer illustrations would greatly improve readability and understanding.

2. The central insight of this paper is that incorporating prompt and sampling diversity at inference time can significantly enhance the reasoning performance of large language models. Nonetheless, this direction is not entirely new: prior studies, such as DIPPER [1], have also highlighted the importance of fidelity and diversity, while [2] demonstrated that modifying prompts to induce greater diversity can further improve ensemble performance.

[1] Lau, Gregory Kang Ruey, et al. "Dipper: Diversity in prompts for producing large language model ensembles in reasoning tasks." arXiv preprint arXiv:2412.15238 (2024).

[2] Naik, Ranjita, et al. "Diversity of thought improves reasoning abilities of large language models." (2023).

**Questions:**

It appears that the task-level perturbations employ a thinker model to generate multiple solutions through sophisticated prompts. I wonder whether incorporating these generated solutions into the raw CoT prompt would result in longer or more elaborate reasoning chains compared to the original CoT reasoning. Could the authors provide corresponding results or statistical analyses on this?

---

> ### Author Response · Authors · 2025-11-24
> **Official response to reviewer kUBb (1/2)**
>
> Dear reviewer kUBb,
>
> Thanks for your concerns and suggestions. We address your concerns below:
>
> **W1. The writing and presentation are weak and difficult to follow. For example, in Section 3 (line 144), the term RNG appears without any prior introduction or explanation. Additionally, the two hypotheses are presented without sufficient depth or illustrative support; although Remark 3.2 attempts clarification, it remains unclear. Including concrete examples or clearer illustrations would greatly improve readability and understanding.**
>
> **Response:**  We thank the reviewer for the feedback on our writing in Section 3.
>
> First, we have replaced “RNG” simply with “randomness” in our revised manuscript. For Hypothesis 3.1 and Hypothesis 3.3, we have enhanced our remarks to include more specific examples and additional justifications, as follows:
>
> **Remark 3.2** (Intuition for Hypothesis 3.1)
>
> Unless $r$ is truly unsolvable for *all* $(\eta,\xi)$, varying the auxiliary randomness $\xi$ induces variation into the success probability. This is because a large auxiliary diversity (e.g. prompt perturbation) can lead to aggressively different decoding paths compared to base policy, which can then lead to larger variations in success probabilities. Note that averaging over $\eta$ reflects that we compare attempts under the same baseline stochasticity.
>
> **Remark 3.4** (Intuition for Hypothesis 3.3)
>
> Here the auxiliary choice $\xi$ is *fixed* on both sides so that both policies enjoy the same amount of base variability ($\eta\sim\nu_0$). Fidelity requires that switching to a new auxiliary setting $\xi^\star$ does not globally worsen the response quality compared to a reference $\xi^0$. It is worth noting that Hypothesis 3.3 should be considered a guideline for selecting good auxiliary diversity sources rather than an assumption: good decoding modes should preserve the fidelity of generation results. For example, with prompt perturbation, one should first test if a specific perturbation method degrades generation quality before applying it as part of the diversity suite.
>
>
>
> **W2.1. The central insight of this paper is that incorporating prompt and sampling diversity at inference time can significantly enhance the reasoning performance of large language models. Nonetheless, this direction is not entirely new**
>
> **Response:** We thank reviewer for concerning about the central insight. However, this paper is **not** about discovering that incorporating prompt and sampling diversity at inference time can significantly enhance LLM reasoning performance; this point has already been stated in the first paragraph of the introduction. Instead, as we stated in the manuscript, we provide **a systematic analysis** of the effect of introducing diversity via perturbations on test-time scaling, including:
>
> - We provide a theoretical foundation analyzing **why** diversity enhanced Best-of-$N$ performance, suggesting that increasing sampling diversity leads to faster convergence as $N\rightarrow \infty$ and a lower asymptotic error. We hope that our theoretical framework will help *shape* and *inspire* future analytical directions for studying scaling inference.
> - Building on our main theorem, we derive a diversity–fidelity trade-off principle that guides the design of sampling strategies. This principle is supported by our empirical observation that moderately meaningful perturbations are particularly effective, while overly similar or irrelevant ones yield no performance gains and can even lead to degradation.
> - We theoretically and empirically analyze **when** diversified exploration remains effective, showing that sampling diversity is effective across a wide range of conditions, but that its benefits depend on many factors, such as the strengths of the policy and thinker models and the perturbation cardinality.
> - Our theoretical proposition shows that majority voting constitutes a failure mode in which diversity does not lead to performance gains, a behavior that is also confirmed empirically, suggesting that diversity cannot be applied indiscriminately.
> - We conduct extensive and systematic evaluations of sampling diversity on reasoning, mathematics, and code generation tasks, quantifying the performance gains it can deliver.
>
> Our central sense is to **provide a systematic analysis that offers a theoretical and empirical foundation for understanding the effect of diversity in LLM inference-time scaling.**
>
> We believe that as a technique develops, **systematic analysis** becomes necessary. Practitioners may know that the technique works, but often lack a clear understanding of **why** it works, **under what conditions** it is effective, and **how** it should be used. As a technique is applied in more scenarios, relying solely on intuition and empirical usage cannot drive its further progress or inspire new techniques. We believe that providing such systematic analysis belongs to what the academic community aim to do.

---

> ### Author Response · Authors · 2025-11-24
> **Official response to reviewer kUBb (2/2)**
>
> **W2.2. prior studies, such as DIPPER [1], have also highlighted the importance of fidelity and diversity, while [2] demonstrated that modifying prompts to induce greater diversity can further improve ensemble performance**
>
> **Response:** Thank you for pointing out the related literature. [R2] was already included in the introduction, in Section 7 (“Additional Related Work”), and in Appendix F (“Extended Discussion of Related Works”).  In the revised manuscript, we have also added [R1] to the first paragraph of the introduction and incorporated it into Section 7.
>
>
>
> [R1] Lau, Gregory Kang Ruey, et al. "Dipper: Diversity in prompts for producing large language model ensembles in reasoning tasks." arXiv preprint arXiv:2412.15238 (2024).
>
> [R2] Naik, Ranjita, et al. "Diversity of thought improves reasoning abilities of large language models." (2023).
>
>
>
> **Q1. It appears that the task-level perturbations employ a thinker model to generate multiple solutions through sophisticated prompts. I wonder whether incorporating these generated solutions into the raw CoT prompt would result in longer or more elaborate reasoning chains compared to the original CoT reasoning. Could the authors provide corresponding results or statistical analyses on this?**
>
> **Response:**  Thanks for your thoughtful question. We evaluated the None and Dual methods using GPT-4o-mini, each combined with a CoT prompt, on HumanEval using Best-of-10 and Best-of-20 sampling. We measured the average number of output tokens of policy model (#Average Output Token) and obtained the following results, based on 3 runs per setting, reporting the mean and std:
>
> |            | CoT-None         | CoT-Dual          |
> | ---------- | ---------------- | ----------------- |
> | Best-of-10 | 1673.3 $\pm$ 6.8 | 1574.0  $\pm$ 6.0 |
> | Best-of-20 | 3345.8 $\pm$ 5.4 | 3164.6 $\pm$ 14.5 |
>
> We observe that the perturbation Dual with CoT prompt does not increase the output length of the CoT reasoning chains compared with the raw CoT prompt.
>
> Thank you for taking the time to review our paper. We hope our responses have addressed your concerns, and we are pleased to respond to any further questions or comments you may have.

---

> > ### Author Response · Authors · 2025-11-26
> > **Kind reminder to Reviewer kUBb (discussion deadline is approaching)**
> >
> > Dear Reviewer kUBb,
> >
> > Thank you again for your time and efforts in reviewing our paper. We have carefully responded to each of your questions.
> >
> > Given that the author-reviewer discussion deadline is approaching, we would greatly appreciate it if you could kindly review our responses and share your valuable feedback. We would be grateful if you could consider increasing the score given our clarification.
> >
> > Thank you very much!
> >
> > Best regards,
> >
> > Authors

---

> ### Author Response · Authors · 2025-11-25
> **A sincere and kind reminder to the ICLR Reviewer kUBb**
>
> Dear ICLR Reviewer kUBb,
>
> We greatly appreciate your time and the insightful comments provided during the review of our paper. We have made extensive efforts to address all your questions, suggestions, and misunderstandings in the response and believe that they adequately address all your concerns. We are grateful for your constructive suggestions.
>
> **With the discussion phase ending soon, we would like to confirm whether there are any other clarifications they would like. If the reviewer's concerns are clarified, we would be grateful if the reviewer could increase the score.**
>
> Thank you again for your time and valuable input; we are deeply appreciative.
>
> Best regards,
>
> Authors

---

### Official Review · Reviewer_1sDt · 2025-11-01

**Soundness:** 3
**Presentation:** 2
**Contribution:** 3
**Rating:** 6
**Confidence:** 4

**Summary:**

This paper presents a sampling strategy that injects an auxiliary diversity source and aims to improve performance, arguing that diversified prompt perturbations enhance Best-of-N inference scaling across reasoning, math, and code tasks. It begins by theoretically justifying its claim, and then empirically demonstrates that the diversified perturbations are effective in different tasks.

**Strengths:**

- It gives a theoretically sound justification of why exploration diversity improves best-of-N.
- The evaluation is comprehensive, which covers the task-level perturbations and query-level perturbations.

**Weaknesses:**

- The presentation could be improved if the notation or symbols could be used more concisely or minimally. And “RNG” is never defined.
- The line 144 is misleading. Unless the success probability of LLM solving one particular problem is defined precisely to be conditioned on the diversity source, it (i.e., the success probability) should be static. I can understand the author’s intention. Maybe the author can define it as a sampling-based success probability where different sampling diversities would have different expectations.
- The hypothesis 3.3 could be unreasonably strong, as it assumes that any auxiliary diversity resources would not make the final success probability sufficiently worse. This is generally not true in practice. A more reasonable assumption could be placed within a finite set of diversity choices, which bounds the worst case.
- Intuitively, the first absolute central moment defined in eq. 2 should be larger (or the constant \hat{\mu}_1 should be larger) if the auxiliary diversity is larger. Could the author show or explain this is always (approximately) true?
- I am aware that there are some relevant works on diversified sampling that are not discussed. Specifically, [1] aims to inject auxiliary diversity into the inference stage to boost performance, where the diversity of the selected instructions is maximized using logdet. [2] balances diversity and risk in LLM sampling based on a prefix tree. These works should at least be discussed or even compared.
- Only cosine similarity metric is adopted as a measure of diversity. I suggest the author could compare with the approach from [1] which use a greedy approach to maximize the diversity.

[1] Hu, W., Lau, G.K., Liu, D., Chen, J., Ng, S., & Low, K.H. (2024). Dipper: Diversity in Prompts for Producing Large Language Model Ensembles in Reasoning tasks. ArXiv, abs/2412.15238.

[2] Zhou, Y., Keuper, M., & Fritz, M. (2025). Balancing Diversity and Risk in LLM Sampling: How to Select Your Method and Parameter for Open-Ended Text Generation. In Proceedings of ACL 2025.

**Questions:**

Please see the weaknesses above.

---

> ### Author Response · Authors · 2025-11-24
> **Official response to reviewer 1sDt (1/2)**
>
> Dear reviewer 1sDt,
>
> We appreciate your insightful comments and provide our response as follows:
>
> **W1. The presentation could be improved if the notation or symbols could be used more concisely or minimally. And “RNG” is never defined.**
>
> **Response:**  We thank the reviewer for pointing this out. We have replaced “RNG” simply with “randomness” in our revised manuscript.
>
> **W2. The line 144 is misleading. Unless the success probability of LLM solving one particular problem is defined precisely to be conditioned on the diversity source, it (i.e., the success probability) should be static. I can understand the author’s intention. Maybe the author can define it as a sampling-based success probability where different sampling diversities would have different expectations.**
>
> **Response:**  We appreciate the reviewer for pointing out this potential confusion. As the reviewer suggests, our intention behind the definition of diversity for $\zeta$ is precisely **sampling-based** diversity at decoding, which provides the only auxiliary randomness on top of the inherent randomness of LLM decoding. This should be the case for both $\eta$ and $\xi$, and so our original wording that $\eta\sim\nu_0$ is the base decoding randomness configuration following baseline policy is not precisely correct. We have changed this sentence to the following:
>
> *We factor the diversified attempt mode as $\zeta=(\eta,\xi)$, where $\eta\sim\nu_0$ is the base randomness within decoding configuration (e.g. a suite of different temperatures to be used during sampling).*
>
> For regular sampling where there is no randomness present beyond the inherent randomness in LLM decoding, $\eta$ should simply be a fixed, null variable. However, we decide to still use this decomposition notation instead of removing $\eta$ because we think the current scheme that isolates and analyzes **additional** randomness is more general.
>
> **W3. The hypothesis 3.3 could be unreasonably strong, as it assumes that any auxiliary diversity resources would not make the final success probability sufficiently worse. This is generally not true in practice. A more reasonable assumption could be placed within a finite set of diversity choices, which bounds the worst case.**
>
> **Response:**  Generally speaking, we agree with the reviewer on this issue. Practically, it is unreasonable to assume that any diversity injection can preserve the quality of LLM generation up to a small degree of deviation $\epsilon$. However, we posit that rather than thinking of Hypothesis 3.3 as a strong assumption on diversity, one should consider it a practical guideline for selecting good diversity sources. We added the following to Remark **3.4**:
>
> *It is worth noting that Hypothesis 3.3 should be considered a practical guideline for selecting good auxiliary diversity sources rather than an assumption: good decoding modes should preserve the fidelity of generation results.*
>
> We also acknowledge that weakening the hypothesis to only hold on a majority of diversified sampling strategies in terms of probability measure does not change the conclusions by much, and makes the hypothesis more reasonable. To keep things simple, however, we decide not to incorporate this change, but instead leave it as a comment in Remark **B.4**.
>
> **W4. Intuitively, the first absolute central moment defined in eq. 2 should be larger (or the constant \hat{\mu}_1 should be larger) if the auxiliary diversity is larger. Could the author show or explain this is always (approximately) true?**
>
> **Response:**  As we understand it, larger auxiliary diversity should lead to more aggressively different decoding paths, which may lead to larger variations in success probabilities. This is of course not a strict mathematical derivation, since it is difficult to define the magnitude of auxiliary diversity in the first place, but we hope this provides an intuition to why larger diversity leads to larger first-moment in $q(r, \eta, \xi)$.

---

> ### Author Response · Authors · 2025-11-24
> **Official response to reviewer 1sDt (2/2)**
>
> **W5. I am aware that there are some relevant works on diversified sampling that are not discussed. Specifically, [1] aims to inject auxiliary diversity into the inference stage to boost performance, where the diversity of the selected instructions is maximized using logdet. [2] balances diversity and risk in LLM sampling based on a prefix tree. These works should at least be discussed or even compared.**
>
> **Response:**  Thank you for your careful reading and for pointing out additional related literature. In the revised manuscript, we have added [R1] to the first paragraph of the introduction, included it in Section 7 (“Additional Related Work”), and further discussed it in Appendix **F** (“Extended Discussion of Related Works”). We have also discussed [R2] in Section 7 (“Additional Related Work”).
>
> [R1].  Hu, W., Lau, G.K., Liu, D., Chen, J., Ng, S., & Low, K.H. (2024). Dipper: Diversity in Prompts for Producing Large Language Model Ensembles in Reasoning tasks. ArXiv, abs/2412.15238.
>
> [R2]. Zhou, Y., Keuper, M., & Fritz, M. (2025). Balancing Diversity and Risk in LLM Sampling: How to Select Your Method and Parameter for Open-Ended Text Generation. In Proceedings of ACL 2025.
>
>
>
> **W6. Only cosine similarity metric is adopted as a measure of diversity. I suggest the author could compare with the approach from [1] which use a greedy approach to maximize the diversity.**
>
> **Response:** Thanks for your constructive suggestion. We aim to study the effect of sampling diversity in test-time scaling, with a primary focus on providing a principled and systematic analysis of sampling diversity. To this end, we develop a theoretical framework for analyzing its effect. Our theory shows that increasing sampling diversity leads to faster convergence and a lower asymptotic error. Building on this framework, we derive a fidelity–diversity trade-off principle that serves as a design guideline for diversified perturbations, and we further characterize the associated success conditions and failure modes.  Although we present a set of instantiations of our theory, our goal is not to propose a state-of-the-art prompting algorithm.
>
> *Dipper* [R1] is a prompt optimization method. It first generates a large pool of prompt candidates, then introduces a prompt selector that performs a greedy search on a development set using an objective that accounts for both semantic entropy and prompt fidelity, yielding a global subset of prompts shared across all examples in the dataset. In contrast, our main focus is on the effect of sampling diversity. Our query-level perturbations are instantiations of our theory; they operate at the problem level and provide problem-specific ideas.  On the MATH dataset, we evaluate the *Single* variant of query-level perturbation and *Dipper* under a Best-of-$50$ setting using GPT-4o-mini. The results EM@50 are as follows:
>
> |       | None | Dipper | Single |
> | ----- | ---- | ------ | ------ |
> | EM@50 | 0.66 | 0.73   | 0.75   |
>
> We find that the diversified sampling *Single* variant achieves higher performance than *Dipper*.  In the revised manuscript, we provide a more detailed discussion in Appendix E.6, "Connection to Prompt Optimization”.
>
>
>
> [R1].  Hu, W., Lau, G.K., Liu, D., Chen, J., Ng, S., & Low, K.H. (2024). Dipper: Diversity in Prompts for Producing Large Language Model Ensembles in Reasoning tasks. ArXiv, abs/2412.15238.
>
> Thank you for your careful and thorough review; we are pleased to address your further concerns and suggestions.

---

> > ### Author Response · Authors · 2025-11-26
> > **Kind reminder to Reviewer 1sDt (discussion deadline is approaching)**
> >
> > Dear Reviewer 1sDt,
> >
> > Thank you again for your time and efforts in reviewing our paper. We have carefully responded to each of your questions.
> >
> > Given that the author-reviewer discussion deadline is approaching, we would greatly appreciate it if you could kindly review our responses and share your valuable feedback. We would be grateful if you could consider increasing the score given our clarification.
> >
> > Thank you very much!
> >
> > Best regards,
> >
> > Authors

---

> ### Author Response · Authors · 2025-11-25
> **A sincere and kind reminder to the ICLR Reviewer 1sDt**
>
> Dear ICLR Reviewer 1sDt,
>
> We greatly appreciate your time and the insightful comments provided during the review of our paper. We have made extensive efforts to address all your questions, suggestions, and misunderstandings in the response and believe that they adequately address all your concerns. We are grateful for your constructive suggestions.
>
> **With the discussion phase ending soon, we would like to confirm whether there are any other clarifications they would like. If the reviewer's concerns are clarified, we would be grateful if the reviewer could increase the score.**
>
> Thank you again for your time and valuable input; we are deeply appreciative.
>
> Best regards,
>
> Authors

---

### Author Response · Authors · 2025-11-24
**Rebuttal by Authors**

### General Response

Dear AC and Reviewers,

We would like to sincerely appreciate the reviewers for their careful reading and highly constructive comments. For clarity and readability of the paper, the following updates have been made:

- Revised the presentation of our contributions and conclusions to better highlight the significance of our work.
- Refined the constructions in Section 4.1 and Section 5.2 to strengthen and clarify the theoretical analysis.
- Added transitions and improved connections between sections and subsections for smoother narrative flow.
- Included additional evaluations and discussions on the effect of model size in Appendix E.4.
- Added a complexity analysis in Appendix D.
- Added evaluations and discussions of the trade-offs between cost and performance in Appendix E.5.
- Added a discussion connecting our work to prompt optimization and included a comparison to a baseline in Appendix E.6.
- Introduced a section on practical protocols in Appendix H.
- Included some missing related works.
- Added a copyright notice for the icon used in the textbox as a footnote on page 10.
- Minor corrections of typos and rephrasing for improved clarity and conciseness.

We hope these updates address the reviewers’ concerns and further clarify our contributions.

Thanks again,

The Authors

---

### Meta-Review · Area_Chair_tbD2 · 2026-01-07

**Summary:**

## Overview

This paper examines how sampling diversity affects inference-time scaling in large language models, focusing on Best-of-N settings. The authors present Theorem 3.5, which formalizes why diverse sampling reduces error rates, and derive a diversity-fidelity tradeoff principle. They evaluate various prompt perturbation strategies across reasoning, math, and code generation tasks, reporting gains of approximately 10\% in EM@100 and Pass@100 metrics.

The submission received scores of 6, 2, 4, and 2$\rightarrow$higher score from four reviewers—a split that reflects genuine disagreement about whether this type of contribution belongs at ICLR.

## Main Concerns

After carefully reading the paper, all reviewer comments, and the authors' responses, I have serious reservations about this work. My concerns align with several points raised by reviewers, particularly regarding novelty and theoretical depth.

### 1. The Core Insights Are Largely Self-Evident

The paper's central claims essentially formalize intuitions that are already well-understood by practitioners:


- **Theorem 3.5: While mathematically rigorous, this result basically says "sampling from more diverse prompts explores more of the solution space, increasing the chance of finding correct answers." The $O(\sqrt{\hat{\mu_1}})$ convergence improvement quantifies this, but doesn't reveal anything counterintuitive.

- **Diversity-fidelity tradeoff (Section 4.1, Figure 2): The finding that "moderately relevant perturbations work best while irrelevant or identical ones don't" is obvious. Nobody would seriously propose using baking recipes to solve math problems. The inverted-U relationship just confirms common sense rather than providing actionable insights.

- **Majority voting failure (Section 5.2): Proposition 5.1 is essentially definitional. Majority voting selects the mode of the response distribution—if that mode isn't the correct answer, majority voting fails. Observing that diversity can harm majority voting follows immediately from first principles.
\end{itemize

The authors defend this as ``systematic analysis'' rather than discovery, but even as analysis, the intellectual contribution feels thin. We're left with a formalization exercise that doesn't meaningfully advance our understanding.

### 2. Weak and Unvalidated Theoretical Assumptions

**Hypothesis 3.1** assumes a uniform lower bound $\hat{\mu_1} > 0$ on the first absolute central moment across all inputs. The authors exclude cases where $p \approx 0$, but provide no empirical evidence that this holds across their test problems. For easy problems or perturbations with minimal effect, this assumption likely breaks down.

**Hypothesis 3.3** requires that any auxiliary diversity source $\xi^*$ satisfies the fidelity constraint in Equation (3). In their rebuttal, the authors reframe this as a "design guideline" rather than an assumption—which undermines the theoretical framework. If Hypothesis 3.3 is just advice for selecting "good" perturbations, then Theorem 3.5 becomes circular: "good perturbations (those satisfying our assumptions) improve performance."

Critically, the paper lacks robustness analysis. What happens when these hypotheses are violated? How sensitive are the results to the magnitudes of $\hat{\mu_1}$ and $\epsilon$? The authors never empirically measure these quantities or verify that observed gains match theoretical predictions.

### 3. No Algorithmic Novelty

The perturbation strategies (Role, Instruction, RandIdeaInj, RandQReph) are straightforward applications of existing prompting techniques. The ``Jabberwocky'' baseline—random nonsense text—serves as a trivial control. The main empirical finding is that task-aligned perturbations outperform irrelevant or identical ones, which follows directly from the (obvious) diversity-fidelity tradeoff.

The paper proposes no new methods. It's an analysis of known techniques that yields predictable results.

### 4. Questionable Fit for ICLR

Reviewer XCHZ captured this well: ``I think the paper makes valuable insights. I will leave it to the AC to decide if such an analysis paper is a good fit for the venue.''

ICLR typically values papers that push the frontier through novel algorithms, architectures, or surprising empirical discoveries. While the CFP lists ``foundation or frontier models'' as a primary area, it doesn't explicitly encourage purely analytical work that formalizes well-known phenomena without providing actionable innovations.

The authors argue that ``systematic analysis becomes necessary'' as techniques mature. That's true in principle, but this particular analysis doesn't meet the bar. The insights are too incremental, the theoretical framework too assumption-laden, and the practical guidance too obvious.

### 5. Persistent Presentation Issues

Despite revisions, the paper remains dense:

- Section 3 introduces heavy notation ($\nu$, $\zeta$, $\eta$, $\xi$, $q(r,\zeta)$, $M_1(r;\nu_0,\Pi)$) that obscures the simple underlying ideas.
- Figures 3 can be improved: hard to read (overlapping lines, unclear color schemes).
- Transitions between theoretical analysis (Section 3), perturbation design (Section 4), and empirical conditions (Section 5) feel abrupt.

### 6. Limited Reviewer Engagement

 This raises concerns: either the review process failed procedurally, or reviewers found the contribution fundamentally insufficient but chose not to elaborate.

## What About Reviewer kUBb's Critique?

Reviewer kUBb's comments were harsh, and the authors correctly note that one cited paper (Naik et al.) was already in the original submission. However, the core criticism—that the central insight is not novel—is valid. The authors respond that their goal is to formalize \emph{why diversity helps, not to discover \emph{that it helps. Fair enough, but the formalization must provide non-obvious insights to justify publication. Here, it doesn't.

## Strengths Worth Acknowledging

To be balanced:

- The paper tackles a timely problem (inference-time scaling).
- Theorem 3.5 provides a rigorous framework with explicit quantitative predictions.
- The empirical evaluation is comprehensive (multiple tasks, models, conditions).
- Appendix H offers practical protocols, though the guidance is fairly obvious.

## Recommendation

I recommend **rejection**. The paper's core insights are largely self-evident, the theoretical assumptions are strong and unvalidated, and there's no algorithmic novelty. While the authors made substantial revisions during rebuttal, the fundamental issues remain.

This is not a bad paper—it's competently executed and addresses a relevant problem. But it doesn't meet the bar for ICLR. The contribution is incremental: a formalization of known intuitions without surprising insights or practical innovations.

## Constructive Suggestions for the Authors

If the authors revise and resubmit elsewhere, I'd suggest:

\begin{enumerate
- **Empirically validate the theory:** Measure $\hat{\mu_1}$ and $\epsilon$ for your perturbations. Show that observed performance gains quantitatively match Theorem 3.5's predictions. This would elevate the work from ``formalization of intuition'' to ``validated predictive framework.''

- **Provide robustness analysis:** Characterize sensitivity to violations of Hypotheses 3.1 and 3.3. Identify a ``fidelity threshold'' $\epsilon^*$ below which diversity becomes harmful.

- **Develop novel methods:** Create a principled approach for automatically generating perturbations that maximize $\hat{\mu_1}$ while minimizing $\epsilon$. The DIPPER comparison is a step in this direction but needs expansion.

- **Simplify presentation:** Lead with intuition, then formalize. Reduce notation in Section 3. Improve figure clarity.

- **Consider alternative venues:** Workshops (e.g., Workshop on Test-Time Scaling) or application-focused conferences may be more appropriate.
\end{enumerate


## Additional Technical Issue

### Citation Error

There is a bibliographic inconsistency that should be corrected:


- Yang et al. (2024b) is cited in the text referring to the Qwen2.5-Math technical report, but the bibliography likely lists it as Yang et al. (2024) without the ``b'' suffix, preventing correct matching. This should be corrected to ensure proper citation resolution.


## Final Thoughts

The authors clearly put significant effort into this work, and their responsiveness during rebuttal was commendable. Reviewer XCHZ's score increase suggests they successfully addressed some technical concerns. However, the fundamental question remains: does this contribution—an analysis that formalizes well-known intuitions without surprising insights or novel methods—belong at a top-tier venue like ICLR?

I don't think it does. The bar for acceptance should be higher.

**Reviewer Concerns:**

see "Summary" text

**Reviewer Scores:**

see "Summary" text

---

### Decision · Program_Chairs · 2026-01-26

Reject